# Towards Backwards-Compatible Data with Confounded Domain Adaptation

**Calvin McCarter**
**mccarter.calvin@gmail.com**

**Reviewed on OpenReview:** **https://openreview.net/forum?id=GSp2WC7q0r**

## Abstract

Most current domain adaptation methods address either covariate shift or label shift, but are not applicable where they occur simultaneously and are confounded with each other. Domain adaptation approaches which do account for such confounding are designed to adapt covariates to optimally predict a particular label whose shift is confounded with covariate shift. In this paper, we instead seek to achieve general-purpose data backwards compatibility. This would allow the adapted covariates to be used for a variety of downstream problems, including on pre-existing prediction models and on data analytics tasks. To do this we consider a modification of generalized label shift (GLS), which we call *confounded shift*. We present a novel framework for this problem, based on minimizing the expected divergence between the source and target conditional distributions, conditioning on possible confounders. Within this framework, we provide concrete implementations using the Gaussian reverse Kullback-Leibler divergence and the maximum mean discrepancy. Finally, we demonstrate our approach on synthetic and real datasets.

## 1 Introduction

The heterogeneity of scientific data is a major obstacle to training useful AI models for scientific research (Bronstein & Naef, 2024). The recent success of AlphaFold (Jumper *et al.*, 2021) for protein structure prediction is in large part due to the availability of a large-scale standardized dataset in the form of the Protein Data Bank (Burley *et al.*, 2019). In contrast, for many other biological problems, data is obtained from a variety of settings which vary due to both scientifically-relevant differences and also due to differences in experimental conditions which give rise to "batch effects" (Leek *et al.*, 2010). And unlike protein structure data, for which ground-truth quantities are absolute distances, many other scientific data modalities report relative quantities (Bronstein & Naef, 2024). These reported quantities that are relative to some detection threshold or noise level are particularly subject to batch effects caused by technical differences in experimental setups, assays, and even computational techniques for processing raw sensor data (Cai *et al.*, 2018).

Within various scientific fields, such as genomics (Johnson *et al.*, 2007; Sprang *et al.*, 2022), proteomics (Gregori *et al.*, 2012; Pelletier *et al.*, 2024), and neuroscience (Yamashita *et al.*, 2019; Torbati *et al.*, 2021), there exist separate preexisting literatures on domain adaptation methods for combining multiple datasets affected by technical artifacts. We cannot hope to discuss all of them in detail, but generally speaking, these methods seek to align datasets so that they have similar feature distributions (Johnson *et al.*, 2007; Shaham *et al.*, 2017) or similar intra-dataset sample-sample relationships (Haghverdi *et al.*, 2018). This objective is dangerous when the to-be-combined datasets not only differ for technical reasons, but also due to variables that are to be scientifically investigated or predicted (Hicks *et al.*, 2018; Zindler *et al.*, 2020; Antonsson & Melsted, 2024). In this case, the feature distributions of the two datasets should not be aligned to be equal, because the difference in feature distributions is *confounded with* a difference in distributions for some scientifically-relevant variable.

In this work, we propose a feature-space domain adaptation method (Pan *et al.*, 2010; Kouw *et al.*, 2016), aiming to provide scientists with general-purpose backwards-compatible data. This backwards compatibility constraint is especially useful in situations where features are used by a downstream model that cannot be updated for organizational or regulatory reasons. In other words, we will try to estimate what the features would have looked like, had they been obtained using the same technical process as the reference dataset. This use-case rules out previous domain adaptation methods (Huang *et al.*, 2006; Zhang *et al.*, 2013; Tachet des Combes *et al.*, 2020) that reweight samples to match distributions, instead of transforming their features. This also rules out the substantial literature (Wang & Mahadevan, 2009; Liu & Tuzel, 2016; Tzeng *et al.*, 2017; Chen *et al.*, 2022) that aims to learn a new (often lower-dimensional) domain-invariant representation.

Our proposed method for feature-space confounded domain adaptation, which we dub *ConDo*, comes with another unusual twist. We make the assumption that the user observes the confounding variables, sidestepping the problem of latent confounders, but only at training time, and not at inference time. This means that the feature-space transformation function must map from one feature-space representation to another, without also taking in the confounding variable as input. This imposition notably rules out ComBat (Johnson *et al.*, 2007) and its many variants (Zhang *et al.*, 2020; Torbati *et al.*, 2021; Radua *et al.*, 2020), which are commonly used for batch correction in genomics, neuroscience, and elsewhere. We motivate and illustrate this key assumption in Section 2.1.

To begin to address this challenge we assume a modification of generalized label shift (GLS) (Tachet des Combes *et al.*, 2020) which we call *confounded shift*. Confounded shift does not assume that the confounding variable(s) are identically distributed in the source and target domains, or that the covariates are identically distributed in the source and target domains. Rather, it assumes that there exists an adaptation from target covariates to source covariates such that the source's conditional distribution of covariates given confounders is equal to that of the adapted-target's conditional distribution. However, we do not assume that the adapted-target's covariates and source's covariates have the same distribution.

In the rest of the paper, we describe and empirically evaluate the ConDo framework for adapting the target to the source, based on minimizing the expected divergence between adapted-target and source conditional distributions, i.e. conditioning on the confounding variables. We show how to compute the expectation with respect to a prior distribution over the confounders, and for the prior we propose an estimator of the product of the source and target confounder distributions. We propose using the Gaussian reverse Kullback–Leibler divergence (KLD) and the maximum mean discrepancy (MMD) as divergence functions. Furthermore, using this framework we provide concrete implementations based on the assumption that the source-vs-target batch effect is "simple". In particular, we restrict the adaptation to be affine, or even location-scale (i.e. with a rotation representable by a diagonal matrix). This assumption is especially intended to adapt structured data, such as biometric sensor outputs, genomic sequencing data, and financial market data, where domain shifts are typically simple, yet where the input-output mapping is often nonlinear. We are not, in this paper, attempting to enable an image classification model for photographs to be adapted to hand drawings, though we hope our framework can be extended to such nonlinear transformations.

## 2 Preliminaries

We begin by providing a concrete motivating example. We then introduce our notation, describe standard approaches to linear domain adaptation, and provide background on generalized label shift.

### 2.1 A motivating example

Suppose you have patient health outcome data and EEG data from the first version of an EEG device, depicted as "V1 training data" in Figure 1. Using this dataset, you have already conducted various statistical analyses and trained health outcome prediction models, including for seizure and depression. But then the EEG machine gets updated to V2, and we obtain a small amount of EEG data collected from the V2 machine, along with patient seizure status, depicted as "V2 training data". The V2 data distribution appears shifted relative to that from the V1 machine. At this point, it seems appropriate to perform covariate shift domain adaptation, to learn an explicit feature-space transformation that adapts the V1 and V2 data to look alike.

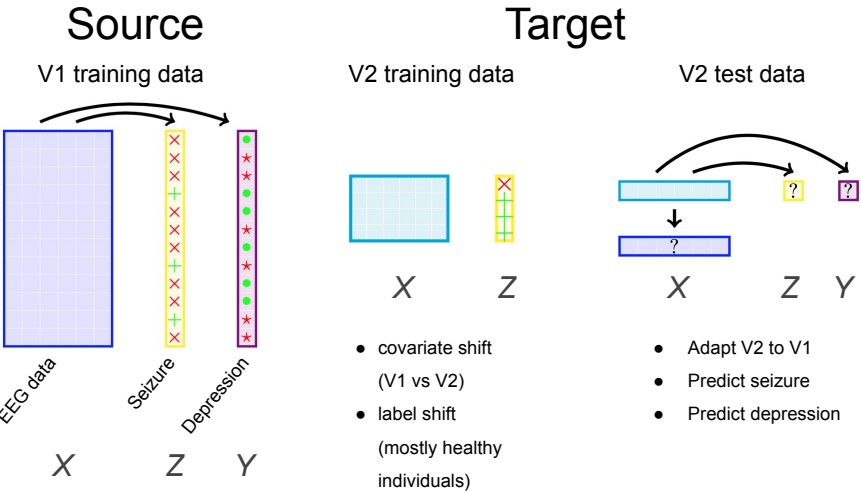

Figure 1: Diagram depicting our motivating scenario. There is confounded shift between source (V1) and target (V2) domains. The different shades of blue of the EEG data portray the "covariate shift" between V1 features and V2 features. The seizure confounding variable differs in distribution between source and target; we portray this "label shift" in seizure status proportions ($\times$ vs $+$) between V1 and V2. With V1 training data, we had previously learned prediction models for seizure and depression given V1 features. With V2 training data, we learn a mapping from V2 features to V1 features via ConDo. At V2 test time, neither seizure nor depression status are provided, but we combine the ConDo V2-to-V1 mapping with the previously-learned prediction models. The EEG data correspond to the features variable $X$, seizure status corresponds to the confounding variable $Z$, and depression status corresponds to the downstream prediction variable $Y$.

Yet additionally, while the V1 dataset comes from a large number of low-risk and high-risk patients, the V2 dataset thus far is mostly comprised of seizure-free individuals. Ignoring the aforementioned covariate shift problem, this latter problem in isolation would fall into the label shift domain adaptation problem setting. Our hypothetical scenario combines these two problems: it has both covariate shift and label shift which are confounded with each other.

Our goal is to learn a feature-space transformation correcting for the technical effects between source (V1 machine) and target (V2 machine) domains, while being aware of the actual neurological differences between the two datasets. We would like to learn a single feature-space transformation (and transformed dataset) that can be used not only for a single prediction task, but for multiple downstream tasks and statistical analyses. For example, we might want to combine the V1 and (adapted) V2 data and then assess for statistical correlation between EEG features and patient age.

The key restriction will be that, at inference time, we will not observe these confounding biological variables. To see why the feature transformation may take in only the EEG data as input, consider the fact that we may want to predict seizure and depression risk on incoming V2 test samples. Even though we could use our confounder (seizure) in the training data to learn a transformation, on test samples we do not yet know health outcomes: indeed, it is what we want to predict! However, we will have access to seizure status while learning the feature transformation, which we will assume fully accounts for the confounding between V1 and V2, as will be formalized in Section 3.

The aforementioned example also illustrates why standard domain adaptation approaches may not be applicable. A new EEG embedding space invariant to the V1-vs-V2 domain shift would not be a "general-purpose" solution for a variety of downstream prediction and statistical inference tasks. In contrast, we would like to preserve as much information as possible, and not only the principal components of variation, or only

the subspaces that are relevant for seizure prediction. Transfer learning (Caruana, 1997; Daume III & Marcu, 2006) and model fine-tuning (Girshick *et al.*, 2014) may also be inapplicable, since the preexisting seizure and depression diagnostic models might be the responsibility of other organizations, or may be non-accessible or non-updatable for regulatory reasons.

## 2.2 Notation

$\mathcal{X}$ and $\mathcal{Z}$ respectively denote the feature (covariate) space and the confounder space. $X$ and $Z$ denote random variables which take values in $\mathcal{X}$ and $\mathcal{Z}$, respectively. A joint distribution over covariate space $\mathcal{X}$ and confounder space $\mathcal{Z}$ is called a domain $\mathcal{D}$. In our setting, there is a source domain $\mathcal{D}_S$ and a target domain $\mathcal{D}_T$; we assume $N_S$ and $N_T$ samples from the source and target domain, respectively. $\mathcal{D}_S^X, \mathcal{D}_T^X$ denote the marginal distributions of covariates under the source and target domains, respectively; $\mathcal{D}_S^Z, \mathcal{D}_T^Z$ denote the corresponding marginal distributions of confounders.

We assume that feature variables are real-valued vectors, so denote samples as $\boldsymbol{x}_S \in \mathbb{R}^{M_S}, \boldsymbol{x}_T \in \mathbb{R}^{M_T}$. We make no such assumption for the confounders, denoting each observation as $z$, but we assume the existence of a user-specified confounder-space kernel function $k_{\mathcal{Z}}(z^{(n_1)}, z^{(n_2)})$. We will seek to optimize the mapping $g : \mathcal{X}_T \to \mathcal{X}_S$ from target feature-space to source feature-space. We will also sometimes consider a target variable (output label) $Y \in \mathcal{Y}$, for which we have a classification hypothesis $h : \mathcal{X} \to \mathcal{Y}$ that has been trained on data from the source domain, corresponding to the standard unsupervised domain adaptation (UDA) setting.

For arbitrary distributions $P$ and $Q$, we assume we have been given a distance or divergence function denoted by $d(P, Q)$. By $\mathcal{N}(\boldsymbol{\mu}, \boldsymbol{\Sigma})$ we denote the Gaussian distribution with mean $\boldsymbol{\mu}$ and covariance $\boldsymbol{\Sigma}$. By $|\cdot|$ we denote the absolute value; by $\det(\cdot)$ we denote the matrix determinant. By $\boldsymbol{A}^\top$ we denote the matrix transpose.

## 2.3 Affine domain adaptation based on Gaussian Optimal Transport

Domain adaptation has a closed-form affine solution in the special case of two multivariate Gaussian distributions. The optimal transport (OT) map under the type-2 Wasserstein metric for $\boldsymbol{x} \sim \mathcal{N}(\boldsymbol{\mu}_S, \boldsymbol{\Sigma}_S)$ to a different Gaussian distribution $\mathcal{N}(\boldsymbol{\mu}_T, \boldsymbol{\Sigma}_T)$ has been shown (Dowson & Landau, 1982; Knott & Smith, 1984) to be the following:

$$\boldsymbol{x} \mapsto \boldsymbol{\mu}_T + \boldsymbol{A}(\boldsymbol{x} - \boldsymbol{\mu}_S) = \boldsymbol{A}\boldsymbol{x} + (\boldsymbol{\mu}_T - \boldsymbol{A}\boldsymbol{\mu}_S), \quad \text{where}$$
$$\boldsymbol{A} = \boldsymbol{\Sigma}_S^{-1/2} \left( \boldsymbol{\Sigma}_S^{1/2} \boldsymbol{\Sigma}_T \boldsymbol{\Sigma}_S^{1/2} \right)^{1/2} \boldsymbol{\Sigma}_S^{-1/2} = \boldsymbol{A}^\top. \tag{1}$$

This mapping has been applied to domain adaptation (Flamary *et al.*, 2019) and other applications (Mallasto & Feragen, 2017; Muzellec & Cuturi, 2018; Shafieezadeh Abadeh *et al.*, 2018; Peyré *et al.*, 2019) in machine learning. For univariate Gaussians $\mathcal{N}(\mu_S, \sigma_S^2)$ and $\mathcal{N}(\mu_T, \sigma_T^2)$, the above transformation simplifies to

$$x \mapsto \mu_T + \frac{\sigma_T}{\sigma_S}(x - \mu_S) = \frac{\sigma_T}{\sigma_S}x + (\mu_T - \frac{\sigma_T}{\sigma_S}\mu_S). \tag{2}$$

## 2.4 Affine domain adaptation minimizing the maximum mean discrepancy

An alternative approach can be derived from representing the distance between target and source distributions as the distance between mean embeddings. This leads to minimizing the (squared) maximum mean discrepancy (MMD), where the MMD is defined by a feature map $\phi$ mapping features $\boldsymbol{x} \in \mathcal{X}$ to a reproducing kernel Hilbert space $\mathcal{H}$. We denote the feature-space kernel corresponding to $\phi$ as $k_{\mathcal{X}}(\boldsymbol{x}^{(n_1)}, \boldsymbol{x}^{(n_2)}) = \langle \phi(\boldsymbol{x}^{(n_1)}), \phi(\boldsymbol{x}^{(n_2)}) \rangle$. Because the feature-space vectors are assumed to be real, MMD-based adaptation methods typically use the radial basis function (RBF) kernel, which leads to the MMD being zero if and only if the distributions are identical.

If the transformation is affine from source to target, the loss can be written as follows:

$$
\begin{aligned}
\mathrm{MMD}^2(\mathcal{D}_T, \mathcal{D}_S) =& \mathbb{E}_{\boldsymbol{x}^{(n_1)}, \boldsymbol{x}^{(n_1)'} \sim \mathcal{D}_T} k_{\mathcal{X}}(\boldsymbol{x}^{(n_1)}, \boldsymbol{x}^{(n_1)'}) \\
& - 2\mathbb{E}_{\boldsymbol{x}^{(n_1)} \sim \mathcal{D}_T, \boldsymbol{x}^{(n_2)} \sim \mathcal{D}_S} k_{\mathcal{X}}(\boldsymbol{x}^{(n_1)}, \boldsymbol{A}\boldsymbol{x}^{(n_2)} + \boldsymbol{b}) \\
& + \mathbb{E}_{\boldsymbol{x}^{(n_2)}, \boldsymbol{x}^{(n_2)'} \sim \mathcal{D}_S} k_{\mathcal{X}}(\boldsymbol{A}\boldsymbol{x}^{(n_2)} + \boldsymbol{b}, \boldsymbol{A}\boldsymbol{x}^{(n_2)'} + \boldsymbol{b}).
\end{aligned}
\tag{3}
$$

Prior work has sometimes instead assumed a location-scale transformation (Zhang *et al.*, 2013), or a nonlinear transformation (Liu *et al.*, 2019a). Notably, while previous MMD-based domain adaptation methods have matched feature distributions (Zhang *et al.*, 2013; Liu *et al.*, 2019a; Singh *et al.*, 2020; Yan *et al.*, 2017), joint distributions of features and label (Long *et al.*, 2013), or the conditional distribution of label given features (Long *et al.*, 2013), they have generally not considered matching the conditional distribution of features given labels. One exception to this is IWCDAN (Tachet des Combes *et al.*, 2020), which however aligns datasets via sample importance weighting rather than a feature-space transformation.

## 2.5 Background on Covariate Shift, Label Shift, and Generalized Label Shift

Domain adaptation methods typically assume either covariate shift or label shift. With covariate shift, the marginal distribution over covariates differs between source and target domains. However, for any particular covariate, the conditional distribution of the label given the covariate is identical between source and target. With label shift, the marginal distribution over labels differs between source and target domains. However, for any particular label, the conditional distribution of the covariates given the label is identical between source and target domain.

More recently, generalized label shift was introduced to allow covariate distributions to differ between source and target domains (Tachet des Combes *et al.*, 2020). Generalized label shift (GLS) instead assumes that, given a transformation function $\tilde{X} := g(X)$ applied to inputs from both source and target domains, the conditional distributions of $\tilde{X}$ given $Y = y$ are identical for all $y$. This is a weak assumption, and it applies to our problem setting as well. However, it is designed for the scenario where we simply need $g$ to preserve information only for predicting $Y$ given $X \sim \mathcal{D}_S^X$.

# 3 Confounded Domain Adaptation

Consider our motivating scenario in which our ultimate goal is to reuse a classification hypothesis $h : \mathcal{X} \to \mathcal{Y}$ in a new deployment setting. We treat the deployment setting as the target domain. And instead of learning an end-to-end predictor for the deployment domain, we learn an adaptation $g$ from it to the source domain for which we have a large number of labeled examples. Then, to perform predictions on the deployment (target) domain, we first adapt them to the source domain, and then we apply the prediction model trained on the source domain. In other words, we do not need to retrain $h$, and instead apply $h \circ g$ to incoming unlabeled target samples. Similarly, other prediction tasks and statistical analyses can be applied after combining adapted-target feature data and source feature data.

In many real-world structured data applications, new data sources are designed with "backwards-compatibility" in mind, with the goal that updated sensor and assays provide at least as much information as the earlier versions. We therefore assume the existence of a "true" noise-free mapping $g$ from the deployment (target) domain to the large labeled dataset (source) domain. The algorithms developed under our framework could instead be applied when treating the deployment setting as the target domain and the labeled dataset as the source domain. However, this easier setting would allow retraining $h$ on adapted data, and is thus not the focus of this paper.

## 3.1 Our Assumption: Confounded Shift

In our case, given $X \sim \mathcal{D}_T^X$, we instead want to recover what it would have been had we observed the same object from the data generating process corresponding to the source domain $X \sim \mathcal{D}_S^X$. In other words, the mapping $g(X)$ should not only preserve information in $X$ useful for predicting $Y$, but ideally all information in $X \sim \mathcal{D}_T^X$ that is contained in $X \sim \mathcal{D}_S^X$.

Table 1: Domain adaptation settings

| Name | Shift | Assumed Invariant |
|------|-------|-------------------|
| Covariate Shift | $\mathcal{D}_S^X \neq \mathcal{D}_T^X$ | $\forall x \in \mathcal{X}, \mathcal{D}_S(Y\|X = x) = \mathcal{D}_T(Y\|X = x)$ |
| Label Shift | $\mathcal{D}_S^Y \neq \mathcal{D}_T^Y$ | $\forall y \in \mathcal{Y}, \mathcal{D}_S(X\|Y = y) = \mathcal{D}_T(X\|Y = y)$ |
| Generalized Label Shift | $\mathcal{D}_S^Y \neq \mathcal{D}_T^Y$ | $\forall y \in \mathcal{Y}, \mathcal{D}_S(g(X)\|Y = y) = \mathcal{D}_T(g(X)\|Y = y)$ |
| **Confounded Shift** | $\mathcal{D}_S^Y \neq \mathcal{D}_T^Y$ | $\forall y \in \mathcal{Y}, \mathcal{D}_S(X\|Y = y) = \mathcal{D}_T(g(X)\|Y = y)$ |

**Graphical representation** We may formulate our setting with latent variables $\tilde{X}$ corresponding to features before the target-to-source mapping. We begin by defining an indicator variable $D$ which specifies whether a sample is taken from the target or the source domain. We may then depict our assumption with the following graphical model

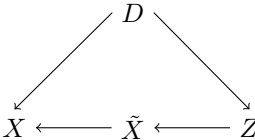

where latent features $\tilde{X}$ always follow the target domain distribution $p_{\mathcal{D}_T}(\tilde{X} = \tilde{x}|Z = z)$, while observed features follow

$$p(X = x|D, \tilde{X} = \tilde{x}) = \begin{cases} \delta(x - \tilde{x}) & D = T \\ \delta(x - g(\tilde{x})) & D = S \end{cases} \tag{4}$$

where $\delta$ is the Dirac delta. By inspection of the graphical model, our setting is a combination of *prior probability shift* and *covariate observation shift* as defined in (Kull & Flach, 2014). Note that latent features $\tilde{X}$ are generated from confounders $Z$, which motivates using a generative model for domain adaptation.

**Relation to Generalized Label Shift** Suppose GLS intermediate representation $g(X)$ were extended to be a function of both $X$ and the source-vs-target indicator variable $D$. Then, given this extended representation $\{X, D\}$, we restrict $\tilde{g}(\{X, D\})$ as follows,

$$\tilde{g}(\{X, D\}) = \begin{cases} g(X) & D = T \\ X & D = S \end{cases} \tag{5}$$

so that samples from the target distribution are adapted by $g(\cdot)$, while those from the source distribution pass through unchanged. For comparability, we let $Y = Z$, i.e. we consider the scenario where our downstream prediction task is to predict the confounding variable which is unavailable at test time. With this extended representation, as well as the restriction on $\tilde{g}$, confounded shift and GLS coincide. The previous assumptions as well as our confounded shift assumption are summarized in Table 1. Note that while confounded shift is stronger than GLS, both allow $\mathcal{D}_S^X \neq \mathcal{D}_T^X$; and just as GLS allows $\mathcal{D}_S^{g(X)} \neq \mathcal{D}_T^{g(X)}$, we analogously allow $\mathcal{D}_S^{g(X)} \neq \mathcal{D}_T^X$.

**Application to downstream prediction tasks** For scenarios where $Y \neq Z$ (where our downstream task is to predict a variable other than the confounder), we do not require dependence or correlation between the confounding variable $Z$ and downstream prediction variable $Y$. Rather, we assume that the conditional distribution of $Y$ given features has not changed from source to target, i.e. $p_{\mathcal{D}_S}(Y|X) = p_{\mathcal{D}_T}(Y|g(X))$. What is necessary is that we correctly map features from target to source; then any downstream prediction model trained on source data can be reused via the composition $h \circ g$, as long as the downstream conditional distribution (estimated by $h$) has not changed.

## 3.2 Main Idea

Our primary aim is to infer a transformation that is broadly applicable, so given observed source domain features $X = x$ we will seek to reconstruct $z$ with minimal error. Our secondary aim is to minimize error

on downstream prediction tasks, which for simplicity is assumed to be binary classification. Formally, the hypothesis is a fixed binary classification function $h : \mathcal{X} \to \{0, 1\}$. We seek to choose $\hat{g}$ which minimizes the accuracy loss induced (by unknown shift $g^{-1}$) on hypothesis $h$ under distribution $\mathcal{D}_T^X$:

$$p_{\mathcal{D}_T^X}\Big(h \circ \hat{g}\big(g^{-1}(X)\big) \neq h(X)\Big). \tag{6}$$

As in typical domain adaptation settings, we expect that $N_T < N_S$, with an abundance of prediction labels on our source dataset (i.e. from the V1 sensor).

Our proposal, which we dub ConDo, is to minimize the expected distance (or divergence) $d$ between the conditional distributions of source and target given confounders, under some specified prior distribution over the confounders:

$$\min_{f_\theta} \mathbb{E}_{z \sim \hat{\mathcal{D}}_Z}\ d\Big(\mathcal{D}_T(\boldsymbol{x}|Z = z), \mathcal{D}_S(f_\theta(\boldsymbol{x})|Z = z)\Big). \tag{7}$$

Four ingredients are needed to turn this into a concrete algorithm: a feature-space transformation $f_\theta : \mathcal{X} \to \mathcal{X}$ (Section 3.3), a prior confounder distribution $\hat{\mathcal{D}}_Z$ (Section 3.4), a sampler from the conditional distributions $\mathcal{D}.(\boldsymbol{x}|Z = z)$ (Section 3.5), and a distance/divergence function $d$ (Section 3.6).

### 3.3 Feature-space transformation function

Our goal is to find the optimal linear transformation $g(\boldsymbol{x}) = \boldsymbol{A}\boldsymbol{x} + \boldsymbol{b}$ of the target to source. In certain scenarios, particularly scientific analyses, it is important for explainability that each $i$th adapted feature $[\boldsymbol{A}\boldsymbol{x}^{(n)} + \boldsymbol{b}]_i$ be derived only from the original feature $[\boldsymbol{x}^{(n)}]_i$. So we will examine both full affine transformations and also transformations where $\boldsymbol{A}$ is restricted to be diagonal $\boldsymbol{A} = \mathrm{diag}(\boldsymbol{a})$. The latter is sometimes referred to as a location-scale adaptation (Zhang *et al.*, 2013); this further requires that the source and target features have the same dimension.

### 3.4 The product prior over confounding variable(s)

Our approach is motivated by the desire to minimize the distance between the conditional distributions $\mathcal{D}_S(\boldsymbol{x}|Z = z)$ and $\mathcal{D}_T(\boldsymbol{x}|Z = z)$ only where we can estimate them both with high accuracy. These conditional distribution estimators may be poor extrapolators, with noisy estimates in low-density regions of $\mathcal{D}_S^Z$ and $\mathcal{D}_T^Z$. This suggests choosing to perform minimization over confounder values that are likely under both source $\mathcal{D}_S^Z$ and target $\mathcal{D}_T^Z$ distributions, which motivates using the product of the two distributions.

We estimate the product of $\mathcal{D}_S^Z$ and $\mathcal{D}_T^Z$ as follows. For efficient sampling, we use a non-parametric estimator of the product prior, with non-negative support over the union of confounder values in the source and target datasets, so that it can be represented as probabilistic weights attached to each sample. We note that empirical distributions may have non-intersecting support, such as if $Z$ is a continuous variable. This motivates smoothing the estimators of $\mathcal{D}_S^Z$ and $\mathcal{D}_T^Z$ before taking their product; this avoids taking the product of two Dirac delta functions, which is undefined.

Given a kernel $k_{\mathcal{Z}}$ over the confounder space, we compute,

$$\hat{\mathcal{D}}_\times^Z := \sum_n^{N_S} \boldsymbol{w}_S^{(n)} \delta(z - Z_S^{(n)}) + \sum_n^{N_T} \boldsymbol{w}_T^{(n)} \delta(z - Z_T^{(n)}), \ \text{where} \tag{8}$$

$$\boldsymbol{w}_S^{(n)} \propto \frac{\sum_{i=1}^{N_S} k_{\mathcal{Z}}(Z_S^{(i)}, Z_S^{(n)})}{\sum_{j=1}^{N_S} \sum_{i=1}^{N_S} k_{\mathcal{Z}}(Z_S^{(i)}, Z_S^{(j)})} \times \frac{\sum_{i=1}^{N_T} k_{\mathcal{Z}}(Z_T^{(i)}, Z_S^{(n)})}{\sum_{j=1}^{N_T} \sum_{i=1}^{N_T} k_{\mathcal{Z}}(Z_T^{(i)}, Z_T^{(j)})} \ \text{and}$$

$$\boldsymbol{w}_T^{(n)} \propto \frac{\sum_{i=1}^{N_S} k_{\mathcal{Z}}(Z_S^{(i)}, Z_T^{(n)})}{\sum_{j=1}^{N_S} \sum_{i=1}^{N_S} k_{\mathcal{Z}}(Z_S^{(i)}, Z_S^{(j)})} \times \frac{\sum_{i=1}^{N_T} k_{\mathcal{Z}}(Z_T^{(i)}, Z_T^{(n)})}{\sum_{j=1}^{N_T} \sum_{i=1}^{N_T} k_{\mathcal{Z}}(Z_T^{(i)}, Z_T^{(j)})}. \tag{9}$$

where $\boldsymbol{w}_S$ and $\boldsymbol{w}_T$ are normalized so $\sum_n \boldsymbol{w}_S^{(n)} + \sum_n \boldsymbol{w}_T^{(n)} = 1$. In practice, we will use an RBF kernel for continuous confounders and a Dirac delta kernel for categorical confounders.

### 3.5 Sampling from the conditional distributions

When the confounding variable is discrete, and each value occurs in multiple examples in our dataset, then we merely sample from these datapoints with replacement. Otherwise (i.e. the confounding variable is continuous), we instead need to sample from conditional generative models for $\mathcal{D}_S(\boldsymbol{x}|Z = z)$ and $\mathcal{D}_S(\boldsymbol{x}|Z = z)$. While diffusion modeling (Sohl-Dickstein *et al.*, 2015) is possible, we found that multiple imputation with chained equations (MICE) (Van Buuren *et al.*, 1999), as implemented in MICE-Forest (Wilson *et al.*, 2022) with LightGBM (Ke *et al.*, 2017), provided better conditional generation results, replicating the experimental results in (Jolicoeur-Martineau *et al.*, 2024). To do this, we concatenate the original dataset and a second copy with all features masked and all confounder(s) unmasked. We then request $K_{\mathcal{X}}$ multiple imputations, which are then used as conditionally-generated features per each observed $Z = z$. This process is performed separately for source and target.

### 3.6 Conditional distribution distance/divergence function

Below, we propose using the reverse-KLD and the MMD in our loss function. Both yield simple, efficient algorithms, including a closed-form solution for the reverse KLD with location-scale adaptation. Note that, as we discuss in Future Work, other divergences are possible within our framework. In particular, optimal transport (OT)-based distances are likely to offer higher accuracy at greater computational expense. However, we limit ourselves to these two divergences for their low computational cost, and to focus on the overall proposed framework rather than the computational challenges and opportunities that arise from combining ConDo with OT.

#### 3.6.1 Reverse Kullback-Leibler divergence under Gaussianity

It can be straightforwardly shown that the linear map Eq. (1) derived from OT leads to adapted data being distributed according to the target distribution. That is, $\boldsymbol{\mu}_P + \boldsymbol{A}(\boldsymbol{x} - \boldsymbol{\mu}_Q) \sim \mathcal{N}(\boldsymbol{\mu}_P, \boldsymbol{\Sigma}_P)$. Therefore, the Gaussian KLD from the target distribution to the adapted source data distribution is minimized to 0, and similarly for the KLD from the adapted source data distribution to the target distribution. This motivates using the Gaussian KLD as a loss function, with either the forward KLD $d(P, Q) \coloneqq d_{KL}(P||Q)$ or reverse KLD $d(P, Q) \coloneqq d_{KL}(Q||P)$. Note that we do not model the features as being Gaussian distributed, but rather that the conditional distribution of each sample's features given confounders as having Gaussian noise.

Whether forward or reverse KLD, minimizing Eq. (7) requires estimating the conditional means and conditional covariances, according to both the source and target domain estimators, evaluated at each $z \sim \hat{\mathcal{D}}_Z$. (If the transformation is location-scale rather than full affine, KLD minimization requires only the conditional variances for each feature.) Given $N$ samples in the prior distribution, each with weight given by $\boldsymbol{w}_n, 1 \leq n \leq N$, let the source and target estimated conditional means be given by $\boldsymbol{\mu}_S^{(n)}, \boldsymbol{\mu}_T^{(n)}$, and the conditional covariances be given by $\boldsymbol{\Sigma}_S^{(n)}, \boldsymbol{\Sigma}_T^{(n)}$, respectively.

For the forward-KLD, this leads to the following objective:

$$
\min_{\boldsymbol{A}, \boldsymbol{b}} 2 \log\left(|\det(\boldsymbol{A})|\right) + \sum_{n=1}^{N} \boldsymbol{w}_n * \left[ \text{tr}\left( \left[\boldsymbol{A}\boldsymbol{\Sigma}_S^{(n)}\boldsymbol{A}^\top\right]^{-1} \boldsymbol{\Sigma}_T^{(n)} \right) \right.
$$
$$
\left. + \left(\boldsymbol{A}\boldsymbol{\mu}_S^{(n)} + \boldsymbol{b} - \boldsymbol{\mu}_T^{(n)}\right)^\top \left[\boldsymbol{A}\boldsymbol{\Sigma}_S^{(n)}\boldsymbol{A}^\top\right]^{-1} \left(\boldsymbol{A}\boldsymbol{\mu}_S^{(n)} + \boldsymbol{b} - \boldsymbol{\mu}_T^{(n)}\right) \right]. \quad (10)
$$

While the forward KLD from target to adapted-source appears to be the natural choice, we instead propose to use the reverse KLD. Due to its computational tractability and well-conditioned behavior, the reverse KLD has found wide use in variational inference (Blei *et al.*, 2017), knowledge distillation (Agarwal *et al.*, 2023), and reinforcement learning (Kappen *et al.*, 2012; Levine, 2018). We will see that it also confers benefits in

domain adaptation. The reverse-KLD leads to the following:

$$\min_{\boldsymbol{A},\boldsymbol{b}} -2\log\left(|\det(\boldsymbol{A})|\right) + \sum_{n=1}^{N} \boldsymbol{w}_n * \left[\text{tr}\left(\boldsymbol{\Sigma}_T^{(n)^{-1}}\boldsymbol{A}\boldsymbol{\Sigma}_S^{(n)}\boldsymbol{A}^\top\right)\right.$$
$$\left. + \left(\boldsymbol{A}\boldsymbol{\mu}_S^{(n)} + \boldsymbol{b} - \boldsymbol{\mu}_T^{(n)}\right)^\top \boldsymbol{\Sigma}_T^{(n)^{-1}}\left(\boldsymbol{A}\boldsymbol{\mu}_S^{(n)} + \boldsymbol{b} - \boldsymbol{\mu}_T^{(n)}\right)\right]. \qquad (11)$$

This is more efficient to optimize, requiring a single matrix inversion per sample, rather than once per sample after each optimization update to $\boldsymbol{A}$. This also minimizes the negative log-abs-determinant of $\boldsymbol{A}$, which functions as a log-barrier away from 0, maintaining the same sign of the determinant across optimization iterations. This is useful, because the linear mapping between two Gaussians is not unique. The reverse-KLD, combined with an initial iterate (e.g. the identity matrix) with a positive determinant, chooses the mapping which preserves rather than reverses the orientation. In contrast, the forward-KLD objective is liable to produce iterates with oscillating signs of $\det(\boldsymbol{A})$.

That the $(-\log|\det(\boldsymbol{A})|)$ term arises naturally out of the reverse-KLD is of potential independent interest. Preventing collapse into trivial solutions is a known problem with MMD-based domain adaptation (Singh *et al.*, 2020; Wu *et al.*, 2021). The reverse-KLD objective may inspire a new regularization penalty for this problem. The log-det heuristic was previously proposed (Fazel *et al.*, 2003) as a smooth concave surrogate for matrix rank minimization, while here it prevents rank collapse.

Furthermore, in the case of a location-scale adaptation, the reverse-KLD can be obtained via a fast exact closed-form solution. Further details are given in Appendix A.

Using the Gaussian KLD requires an estimate of the mean and covariance of the features given each value $z$ of the confounder. We obtain this by generating $K_{\mathcal{X}}$ samples of $\boldsymbol{x} \sim \mathcal{D}_S(\cdot|Z=z)$ and $\boldsymbol{x} \sim \mathcal{D}_T(\cdot|Z=z)$ per each $z$, then computing the empirical means and covariances. We optimize the ConDo-KLD objective with the PyTorch-minimize (Feinman, 2021) implementation of the trust-region Newton conjugate gradient method (Lin & Jorge, 1999).

### 3.6.2 The conditional maximum mean discrepancy

We extend MMD-based domain adaptation to match conditional distributions by sampling from the prior confounder distribution. For a particular $z \in \mathcal{Z}$ sampled from the prior, suppose we have a way of sampling from $\mathcal{D}_T(\boldsymbol{x}|Z=z)$ and $\mathcal{D}_S(\boldsymbol{x}|Z=z)$. Then, we have

$$d\left(\mathcal{D}_T(\cdot|Z=z), \mathcal{D}_S(\cdot|Z=z)\right) := \text{MMD}^2(\mathcal{D}_T(\cdot|Z=z), \mathcal{D}_S(\cdot|Z=z)) \qquad (12)$$
$$= \mathbb{E}_{\boldsymbol{x}^{(n_1)}, \boldsymbol{x}^{(n_1)'} \sim \mathcal{D}_T(\cdot|Z=z)} k_{\mathcal{X}}(\boldsymbol{x}^{(n_1)}, \boldsymbol{x}^{(n_1)'})$$
$$- 2\mathbb{E}_{\boldsymbol{x}^{(n_1)} \sim \mathcal{D}_T(\cdot|Z=z), \boldsymbol{x}^{(n_2)} \sim \mathcal{D}_S(\cdot|Z=z)} k_{\mathcal{X}}(\boldsymbol{x}^{(n_1)}, \boldsymbol{A}\boldsymbol{x}^{(n_2)} + \boldsymbol{b})$$
$$+ \mathbb{E}_{\boldsymbol{x}^{(n_2)}, \boldsymbol{x}^{(n_2)'} \sim \mathcal{D}_S(\cdot|Z=z)} k_{\mathcal{X}}(\boldsymbol{A}\boldsymbol{x}^{(n_2)} + \boldsymbol{b}, \boldsymbol{A}\boldsymbol{x}^{(n_2)'} + \boldsymbol{b}). \qquad (13)$$

For the feature-space kernel $k_{\mathcal{X}}$, we use the RBF kernel by default. We minimize this objective with stochastic optimization. For each minibatch, we sample $K_{\mathcal{Z}}$ values from the confounder prior; for each confounder value we sample $K_{\mathcal{X}}$ feature-vectors from each of $\mathcal{D}_T(\boldsymbol{x}|Z=z)$ and $\mathcal{D}_S(\boldsymbol{x}|Z=z)$.

**Additional implementation details** Further implementation details and computational complexity analysis are provided in Appendix B. Our software, with a Scikit-learn (Pedregosa *et al.*, 2011) compatible API and with experimental scripts, is available at `https://github.com/calvinmccarter/condo-adapter`.

## 4 Experiments

We compare ConDo to baseline methods on three synthetic, two hybrid, and three real data settings. When available, we evaluate against the latent ground-truth features by measuring the root-mean-squared error

(rMSE) between these and the adapted features. We also compare the different methods on unsupervised domain adaptation (UDA), measuring the performance of source-trained classifiers on adapted target test data, to evaluate our ability to adapt data for use by non-retrainable downstream models. For these experiments, we employ TabPFN (Hollmann *et al.*, 2022), a state-of-the-art pretrained Transformer-based tabular classifier.

We optimize the MMD and ConDo-MMD objectives with AdamW (Loshchilov & Hutter, 2017), with weight decay $= 10^{-4}$, $\beta_1 = 0.9$, and $\beta_2 = 0.999$. We set $K_{\mathcal{X}} = 20$ (for ConDo-KLD and ConDo-MMD) and $K_{\mathcal{Z}} = 8$ (for ConDo-MMD); for MMD, each step also uses 8 evaluations of the MMD loss with 20 samples. We run for 5 epochs (with an early stopping patience of 3 epochs) with learning rate $= 10^{-3}$, unless otherwise indicated in the Appendix; for both MMD and ConDo-MMD, an epoch is defined as $\min(N_S, N_T)/8$ steps.

### 4.1 Synthetic 1d features with 1d continuous confounder

We first examine confounded domain adaptation in the context of a single-dimensional feature confounded by a 1d continuous confounder. We analyze the performance of vanilla and ConDo adaptations, when the effect of the continuous confounder is linear homoscedastic (left column), linear heteroscedastic (middle column), and nonlinear heteroscedastic (right column). In each case there is confounded shift, because confounder is distributed as Uniform$[0, 8]$ in the target domain, and as Uniform$[4, 8]$ in the source domain. This setting is illustrated in the top row of subplots in Figure 2.

For each of the the three settings, we apply the baseline linear domain adaptation methods, Gaussian OT and MMD, and our ConDo versions of these methods, ConDo Gaussian KLD and ConDo MMD. We then evaluate the different methods via root-mean-squared-error (rMSE) on a heldout dataset of 100 samples, comparing the known ground-truth to the estimates from each adaptation method. Note that in this and all following settings in the rest of the paper, neither the baseline nor the ConDo methods have access to the confounding variable values of the heldout test samples. The results are shown on each of the remaining rows of subplots in Figure 2.

We repeat the above experimental setup, but with modifications to verify whether our approach can be accurate even when its assumptions no longer apply. We run experiments with-and-without label shift (i.e. different distributions over the confounder between source and target), with-and-without feature shift (i.e. with and without batch effect), and with-and-without additional iid $\mathcal{N}(0, 1)$ noise, for a total of 8 settings. Heldout test errors are shown in Figure 3. ConDo strongly outperforms vanilla adaptation whenever there is target shift, and is non-inferior otherwise. In Figure S1, we also show that ConDo offers improved estimation of the feature-space mapping parameters whenever there is target shift, and is non-inferior otherwise.

### 4.2 Synthetic 1d features with multi-dimensional continuous confounders

We extend the previous experiment to consider the scalability of ConDo to multidimensional confounders. For the noise-free, label-shift, feature-shift setting, keeping all other experimental settings and hyperparameters identical, we vary the number of confounders from 1 to 32. We first augment the number of confounders by appending additional irrelevant "confounders", sampled from $\mathcal{N}(0, 1)$, to our inputs to the ConDo method. The results, shown in Figure 4(A), indicate that ConDo is very robust to the inclusion of a moderate number of non-confounders.

We next augment the number of confounders by generating a noisy additive decomposition of our original confounder. We first uniformly sample the "true" confounder as before, and generate the feature from it as before. We then generate a random multidimensional confounder summing to the "true" confounder of the desired dimensionality (Dickinson, 2010), and provide this to the ConDo methods. The results, provided in Figure 4(B), show that ConDo is still effective in spite of this subtle relationship between the feature and the multi-dimensional confounders.

We furthermore use this latter setting to examine the behavior of ConDo when confounders are partially observed. Results, shown in Figure S4, indicate that typically ConDo fails to take advantage of partially-observed confounders while remaining non-inferior to baselines.

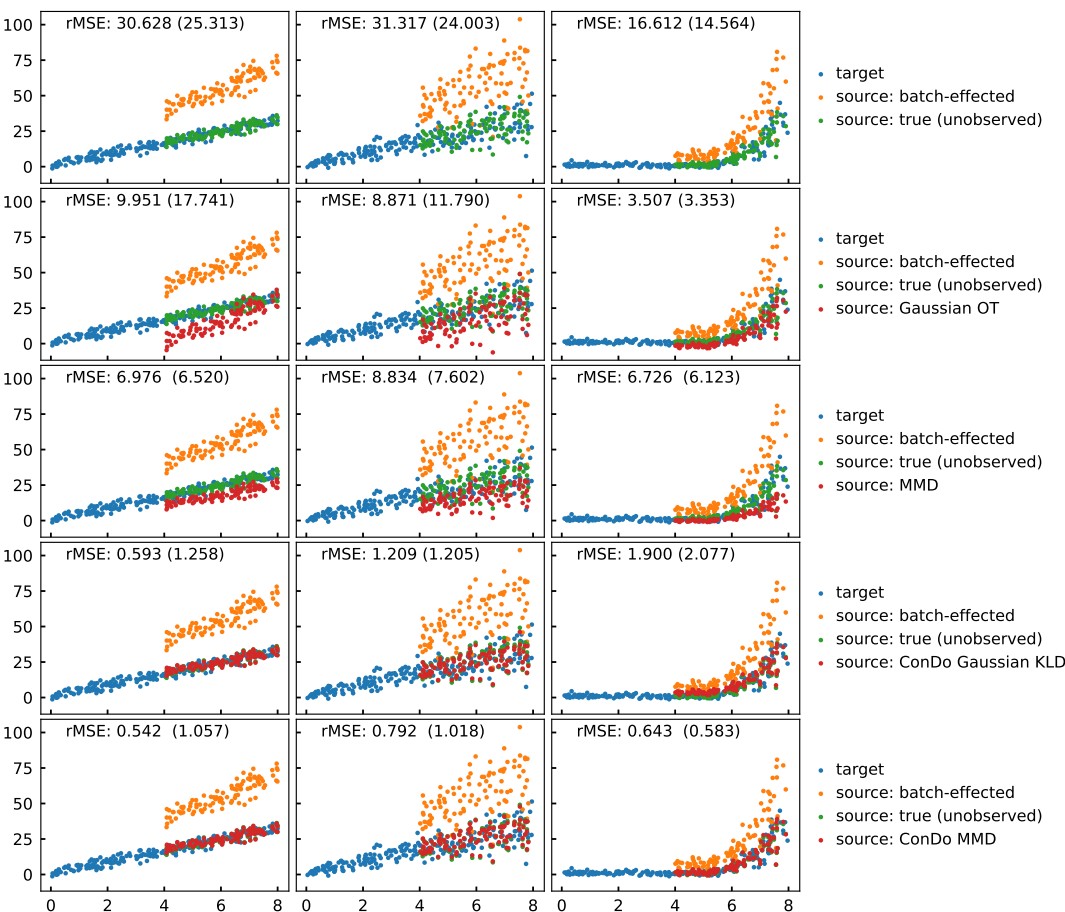

Figure 2: ConDo methods are superior to Gaussian OT when confounded label shift and feature shift are present. The columns, in order, correspond to a confounder with a linear homoscedastic effect, a confounder with a linear heteroscedastic effect, and a confounder with a nonlinear heteroscedastic effect. The first row depicts the problem setup, while the remaining rows depict the performance of Gaussian OT and our ConDo methods. Red points overlapping with green points is indicative of high accuracy. In each subplot, we provide the rMSE on training source data (depicted), and in parentheses, the rMSE on heldout source data (not depicted) generated with confounder sampled from target prior $\mathcal{D}_T^Z$. The printed rMSEs are averaged over 10 independent random simulation runs, while the plots depict the results from the final simulation run.

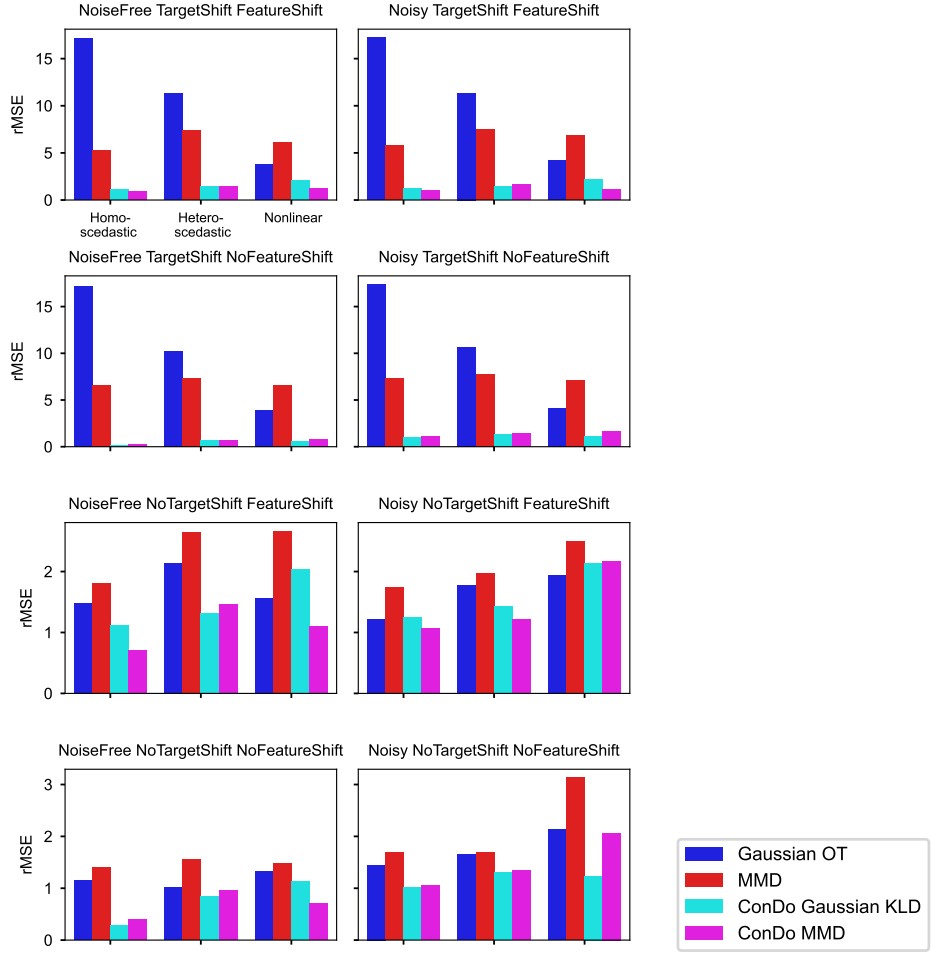

Figure 3: Test errors for experiment with synthetic 1d features with 1d continuous confounder. For each adaptation method, we compute the rMSE of true target feature values vs inferred target feature values after adaptation, then average over 10 simulations.

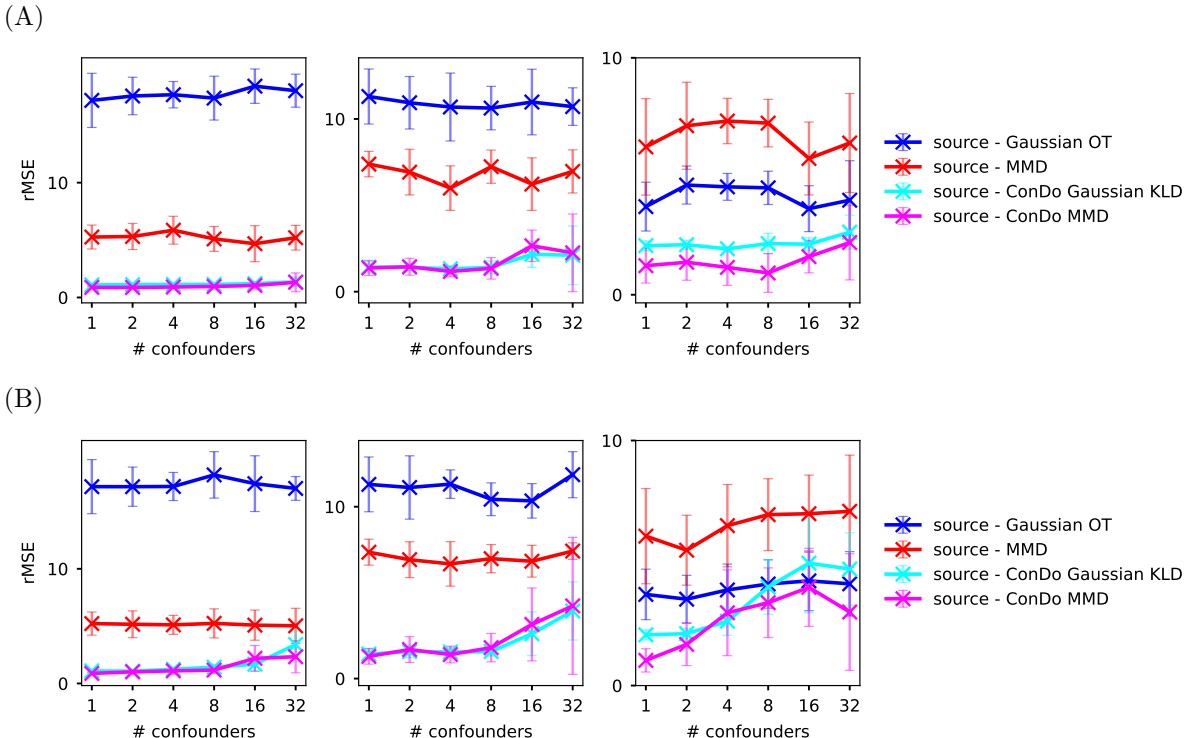

Figure 4: Results for transforming 1d data with multiple continuous confounders, with extra irrelevant $\mathcal{N}(0, 1)$ confounders, shown in (A), and with noisy additive decomposition, shown in (B). The rMSEs are averaged over 10 random simulations are shown for heldout test data (100 samples per simulation). The columns, in order, correspond to a confounder with a linear homoscedastic effect, a confounder with a linear heteroscedastic effect, and a confounder with a nonlinear heteroscedastic effect.

### 4.3 Synthetic 1d and 2d features with 1d categorical confounder

Here, we generate 1d features based on the value of a 1d binary confounder. The source distribution is a mixture of Gaussians $0.25\mathcal{N}(5, 1^2) + 0.75\mathcal{N}(0, 2^2)$, while the target is $0.75\mathcal{N}(5, 1^2) + 0.25\mathcal{N}(0, 2^2)$, and the confounder variable indicates the true mixture component. We also use this setting to analyze the performance of ConDo for a variety of sample sizes. For each sample size under consideration, we run 10 random simulations, and report the rMSE compared to the actual pre-feature shift values.

Results are shown in Figure 5. In Figure 5(A) we see that the ConDo methods outperform the baselines, and that they are robust to small sample sizes. As the number of samples increases, ConDo MMD converges to the correct transformation. We see in Figure 5(B) that ConDo Gaussian KLD is not as fast as the closed form Gaussian OT solution, but still scales nicely with sample size; meanwhile MMD and ConDo MMD have comparable runtimes.

The raw data and results are plotted in Figure S5 in the Appendix. In Figure S6, we also show that ConDo leads to lower-error estimates of the true feature-space mapping. We extend the previous experiment to 2d features, showing reduced rMSE and improved classification accuracy in Figure S7, and improved feature shift parameter estimation in Figure S8.

### 4.4 ANSUR II anthropometric survey data

We next evaluated approaches on the ANSUR II (Gordon *et al.*, 2014) dataset, comprising 93 anthropometric measurements spanning the gamut from ankle circumference to wrist height of 6068 military personnel. We then synthetically created source and target datasets with known ground-truth feature-space transformations

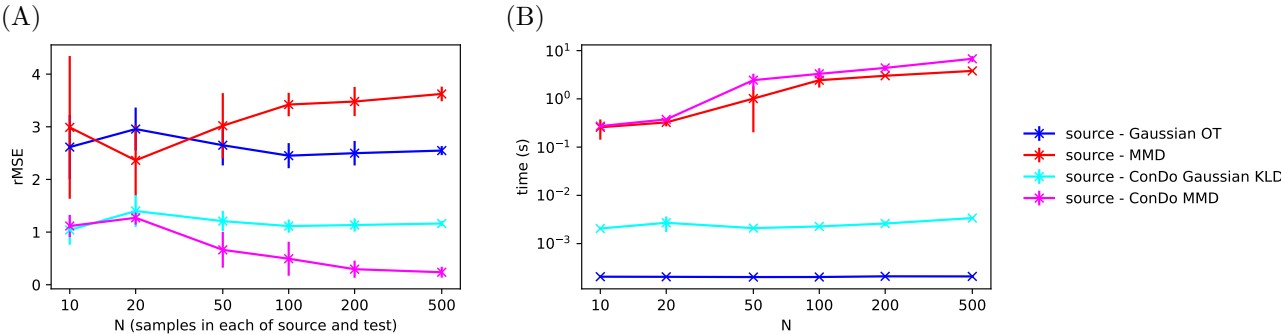

Figure 5: Results for transforming 1d data with a 1d categorical confounder. (A) Plot of rMSE vs sample size for each of the domain adaptation methods. Each rMSE was averaged over 10 simulations, with the vertical lines indicating 1 standard deviation over the simulations. (B) Plot of runtime vs sample size for each of the domain adaptation methods.

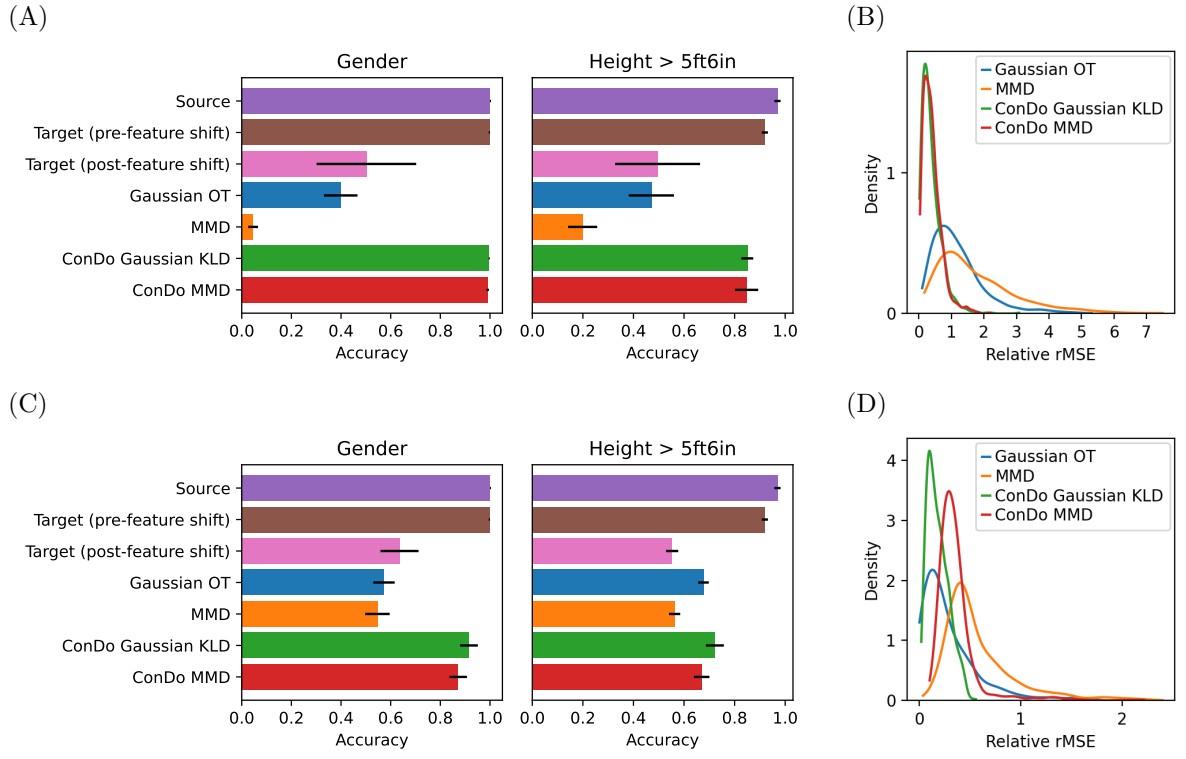

Figure 6: Results on ANSUR II for affine perturbation (A-B) and location-scale perturbation (C-D). Model accuracies depict the average over 10 simulations, with error bars depicting the standard deviation. Kernel density plots in (B,D) depict the density of pre-adaptation rMSE divided by post-adaptation rMSE, averaged over simulations, measured against ground truth data. We see that ConDo is less likely to produce worsened data (i.e. relative rMSE > 1).

as follows. We generated the source (and the target) dataset as a random subsample of 500 individuals with a 75%-25% (and a 25%-75%) male-female split, with gender as the confounding variable. Then, we linearly perturbed the target dataset, as either a random affine or random location-scale transform. For random affine, we use random positive-determinant matrix $\boldsymbol{A} = \boldsymbol{U}\text{diag}(\boldsymbol{d})\boldsymbol{V}^\top$, where $\boldsymbol{U}, \boldsymbol{V}$ are each Haar-distributed orthogonal matrices and $\boldsymbol{d}_i \sim \text{Unif}[0.5, 2]$; for random location-scale, $\boldsymbol{A}_{ii} \sim \text{Unif}[0.5, 2], \boldsymbol{b}_i \sim \text{Unif}[0.5, 2]$. We then trained TabPFN (Hollmann *et al.*, 2022) models for `MaleOrFemale` and `Height > 5ft6in` (the median height) on source data. We applied domain adaptation methods, with ConDo methods having access to male-vs-female as the confounding variable. Prediction models are then applied on adapted target-to-source features. In Figure 6, we show accuracies of the prediction models and rMSEs for the target dataset features, from 10 independent random simulations. We see that ConDo improves upon vanilla adaptation, with ConDo Gaussian KLD providing the best performance across all metrics.

In the Appendix Section C.4, we include results in additional settings. Similar performance is shown for when TabPFN prediction models are instead trained on adapted source-to-target features in Figure S9. We do not observe systematic difference due to ConDo in feature-space mapping parameter estimation, shown in Figure S10. Furthermore, results for only shift in confounder (gender) distribution in Figure S12, for only feature shift in Figure S11, and for neither shift in Figure S13 are also provided, showing improvement, noninferiority, and noninferiority, respectively, for our approach.

### 4.5 Image color adaptation

We here apply domain adaptation to the problem of image color adaptation, treating each image as a dataset with 3 features (for the RGB components). We start by adapting back and forth between two ocean pictures taken during the daytime and sunset (the Python Optimal Transport library (Flamary *et al.*, 2021) Gaussian OT example), depicted in the top row of Figure 7(A). In this scenario, there is no confounding, since the images contains water and sky in equal proportions. Thus, conditioning on each pixel label (a categorical confounder, taking on a value of either "water" or "sky"), is expected to be unnecessary. Below the top row, we show the results of affine adaptation for each of the different methods. On the left column of Figure 7(A), we show the result of applying the learned source-to-target transform. On the right column of Figure 7(A), compute the inverse of the aforementioned source-to-target transform, and treat it as a target-to-source transform. We see that both Gaussian OT and Condo Gaussian KLD learn adaptations that work properly on both source-to-target and target-to-source. MMD produces images with appropriate color balances, but with colors applied to inappropriate parts of the image. ConDo MMD produces a good source-to-target image, but its inverse mapping is bad.

Next, we attempted color adaptation between the ocean daytime photo and another sunset photo including beach, water, and sky, shown in Figure 7(B). Here, there is confounded shift, so ConDo utilizes pixels labeled as "sky", "water", or "sand". Now the source-to-target adapted images have poor (grayish) color balances for the baseline methods, while they have good color balances for the ConDo methods. For the inverse target-to-source mapping, both MMD and ConDo MMD struggle.

More results and details are given in the Appendix Section C.5.

### 4.6 California housing price prediction

Here, we apply ConDo to an unsupervised domain adaptation setting derived from the California housing dataset (Pace & Barry, 1997). We split the data into source and target domains based on the first feature, `Median Income`, defining the source domain as being the housing districts with income less than or equal to the median. The geographical results of this source-target split are shown in Figure 8(A), with examples plotted according to their LatLon features. We see that higher-income districts are geographically clustered, and thus use LatLon coordinates as the two confounding variables. As our classification task, we predict whether the mean house value in each district exceeds the median over the entire dataset, 1.797 (in $100k). As features, we use all remaining features aside from `Median Income`.

We train a TabPFN classifier on the source domain, and evaluate it on target domain, repeated over 10 random simulations, each with 500 source training samples and 500 target test samples in each simulation.

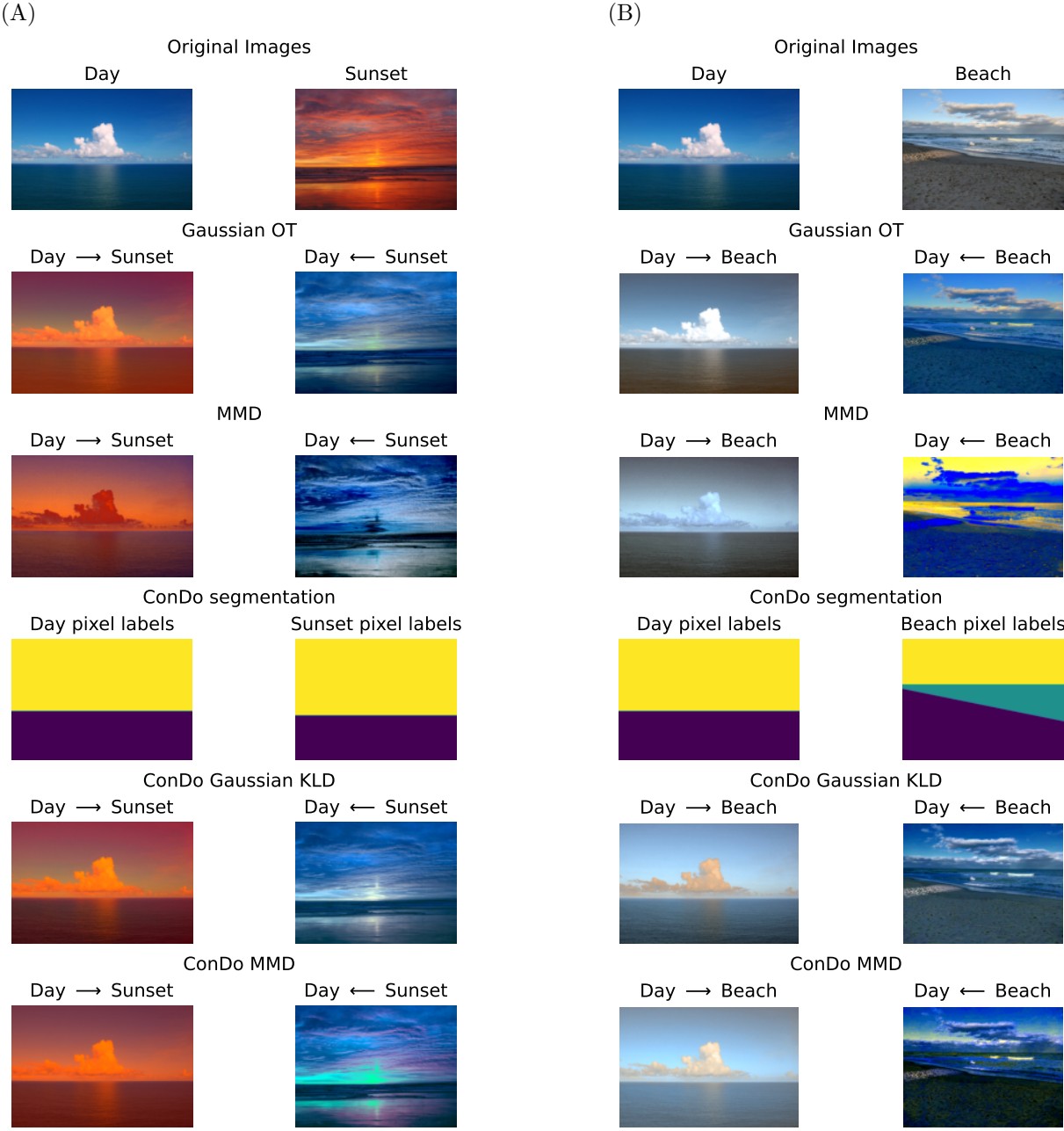

Figure 7: Image color adaptation results without (A) and with (B) confounded shift. We see that ConDo is non-inferior in (A). In (B), we see that non-ConDo methods produce gray-ish Day → Beach images. Meanwhile, ConDo methods produces light blue sky and yellow clouds in Day → Beach images. ConDo Gaussian KLD is the only method to produce normal-looking images for all four tasks.

We perform location-scale domain adaptation to make the target features resemble the source features before applying the classifier. As shown in Figure 8(B-C), vanilla Gaussian OT and MMD methods make the performance even worse than no adaptation. We also see that the ConDo methods better than the vanilla methods, and that ConDo Gaussian KLD improves upon no adaptation. In the Appendix Section C.6, we show similar results for training a classifier on adapted-source-to-target data, with ConDo outperforming the baselines.

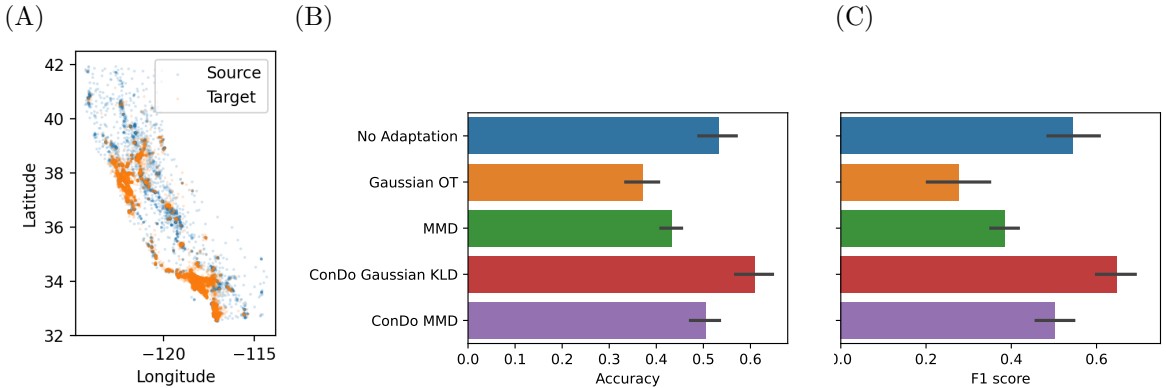

Figure 8: Results on CA Housing. (A) Source-target split based on median income. Accuracy (B) and F1-score (C) on target test data, over 10 random simulations. Error bars depict the standard deviation over simulations.

### 4.7  SNAREseq single-cell multi-omics dataset

We apply domain adaptation to a SNAREseq single-cell dual-omics dataset (Demetci *et al.*, 2022). In the dataset, 1047 cells are co-assayed with RNA-seq for gene expression and ATAC-seq for chromatin accessibility. Each sequenced cell is associated with one of four cell types; unique cell identities are also given, so matched sample pairs across the two modalities are known. We treat ATAC-seq assay features as source domain and RNA-seq assay features as target domain. For ConDo, we perform adaptation controlling for cell type, the confounding variable.

We first simulate unsupervised domain adaptation in which we perform cell type classification with TabPFN (Hollmann *et al.*, 2022). In each simulation, we have a large source domain training set of 500 source cells. We also simulate having a small set of $C$ cells for which we have both source and target domain data available to use for adaptation; we simulate $C \in \{5, 10, 20, 50, 100\}$. We then evaluate the classifier on the remaining unseen cells using target domain data (ATAC-seq features). Because the feature dimensionality differs for the two assays, we only perform MMD and ConDo MMD adaptation with affine transforms. Results for 10 independent random simulations, depicted in Figure 9, show that ConDo MMD outperforms MMD. Similar advantage for ConDo is shown when classifiers were instead trained on adapted source-to-target features in Figure S16 in Appendix Section C.7.

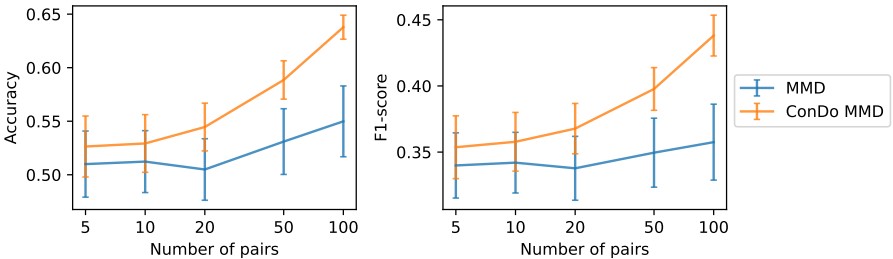

Figure 9: SNAREseq cell type classifier performance on test set. Classifier performance (accuracy and F1-score) is shown as a function of the number of paired samples available to the domain adaptation methods. The error bars depict the standard error computed over 10 random simulations.

Next, we evaluate whether domain adaptation methods produce data showing desirable clustering patterns. We concatenate the adapted-source and target datasets, and compute the silhouette score (the mean of the silhouette coefficients over all samples). This silhouette score calculation requires both features and a label for each sample; we compute silhouette score separately for labels defined as the assay, the cell type, and the

cell identity. A silhouette score approaching +1 indicates that samples have small distance to other points in the same group (i.e. with the same label), compared to points in the next nearest group (i.e. with the smallest average distance). A score of 0 indicates overlapping groups, while a score of -1 indicates discordance between the features and labeling. We desire large scores for cell identity labels and cell type labels, and a score close to 0 for assay labels. ConDo improves upon the baseline for all three settings, as shown in Table 2.

Table 2: Silhouette scores after combining the adapted ATAC-seq and the RNA-seq data into a single dataset with $2 * 1047$ samples. Silhouette scores are computed for different labelings, given in the column headings. "Cell identity" denotes the pairs of samples from the same cell for two assays; "cell type" denotes samples grouped by cell type; "assay" denotes samples being labeled as either ATAC-seq or RNA-seq.

| Method | Cell identity ($\uparrow$) | Cell type ($\uparrow$) | Assay ($\rightarrow 0 \leftarrow$) |
|---|---|---|---|
| MMD | -0.4556 | 0.4574 | 0.0379 |
| ConDo MMD | **-0.4425** | **0.4940** | **0.0218** |

### 4.8 Gene expression microarray batch effect correction

We analyze performance on the *bladderbatch* gene expression dataset commonly used to benchmark batch correction methods (Dyrskjøt *et al.*, 2004; Leek, 2016). We use all 22,283 gene expressions (i.e. features) from this bladder tissue Affymetrix microarray dataset; *bladderbatch* was preprocessed so that each feature is approximately Gaussian. We attempt a location-scale transform, as is typical with gene expression batch effect correction. We choose the second largest batch (batch 2, with 4 cancer samples out of 18 total) as the source domain, and the largest batch (batch 5, with 5 cancer samples out of 19 total) as the target domain. The confounder is 1d categorical (cancer or non-cancer).

For each method, we compute the silhouette scores of the adapted datasets, with respect to the batch variable (and, in parentheses, the test result variable). We desire the silhouette score to be large for the cancer status, and to be close to 0 for the batch label. Because the cancer fractions are roughly the same (4/18 vs 5/19) for batches 2 and 5, we do not expect to need to account for confounding. Results are shown in Table 3. We see that all domain adaptation methods improve upon no adaptation. For cancer status, MMD has the largest silhouette score as desired. For the batch variable, all adaptations produced slightly negative scores. While this is less concerning than the larger magnitude positive value (0.0884) for no adaptation, it may suggest overfitting; the silhouette score is closest to zero for ConDo MMD.

We repeat the experiment after inducing confounding by removing samples. For 10 random simulations, we remove half (7) of the non-cancer samples in batch 2, so that batch 2 is 4/11 non-cancerous, while batch 5 remains 5/19 non-cancerous. Results are shown in Table 4. We see that ConDo methods outperform their baseline counterparts, and that ConDo Gaussian KLD in particular produces gene expression data in which variation concords with cancer status rather than batch.

Table 3: Silhouette scores on *bladderbatch* dataset, on entire dataset without induced confounding. Silhouette scores are computed for different labelings, given in the column headings.

| Method | Cancer Status ($\uparrow$) | Batch ($\rightarrow 0 \leftarrow$) |
|---|---|---|
| No Adaptation | 0.2798 | 0.0884 |
| Gaussian OT | 0.3008 | -0.0279 |
| MMD | **0.3123** | -0.0214 |
| ConDo Gaussian KLD | 0.3121 | -0.0195 |
| ConDo MMD | 0.3107 | **-0.0176** |

Table 4: Silhouette scores on *bladderbatch* dataset, after inducing confounding in the data. Silhouette scores are computed for different labelings, given in the column headings. We also include the standard errors computed over 10 random simulations.

| Method | Cancer Status ($\uparrow$) | Batch ($\rightarrow 0 \leftarrow$) |
|---|---|---|
| No Adaptation | $0.2984 \pm 0.0009$ | $0.0982 \pm 0.0010$ |
| Gaussian OT | $0.2934 \pm 0.0006$ | $-0.0329 \pm 0.0001$ |
| MMD | $0.2966 \pm 0.0010$ | $-0.0372 \pm 0.0001$ |
| ConDo Gaussian KLD | $\mathbf{0.3212} \pm 0.0007$ | $\mathbf{-0.0276} \pm 0.0002$ |
| ConDo MMD | $0.3196 \pm 0.0008$ | $-0.0284 \pm 0.0002$ |

## 5 Related Work

As far as we are aware, previous work on domain adaptation does not describe or address our exact problem. There is a large body of research in domain adaptation which maps both source and target distributions to a new latent representation where they match (Baktashmotlagh *et al.*, 2013; Yan *et al.*, 2017; Ganin *et al.*, 2016; Gong *et al.*, 2016). These however cannot achieve data backwards-compatibility, because they create a new latent domain. Other domain adaptation methods are also inapplicable to our setting since they match distributions via reweighting samples (Cortes & Mohri, 2011; Tachet des Combes *et al.*, 2020) or dropping features (Kouw *et al.*, 2016).

Prior research exists for performing domain adaptation when both features and label are shifted, including the generalized label shift (GLS) / generalized target shift (GeTarS) (Zhang *et al.*, 2013; Rakotomamonjy *et al.*, 2020; Tachet des Combes *et al.*, 2020). However, these methods assume the specific prediction setting where the label is the confounder, and optimize composite objectives that combine distribution matching and prediction accuracy. In our case, the confounder may not be the label of our prediction model of interest, and indeed we may not even be mapping covariates for the purpose of any downstream prediction task. Furthermore, by conditioning on confounders, our framework can handle multivariate confounders or even complex objects which are accessed only via kernels. Landeiro *et al.* introduced the term *confounding shift* to describe a form of GLS/GeTarS, but it does not match our *confounded shift* assumption, since the confounding variables are unobserved. Their method, which comprises confounder detection and adversarial confounder-robust classification, is substantially different from our approach.

The most relevant domain adaptation methods for our context perform asymmetric feature transformation, in which source features are adapted to target features, and are thus compatible with general-purpose backwards compatibility. EasyAdapt (Daumé III, 2007) and EasyAdapt++ (Daumé III *et al.*, 2010) are notably successful approaches performing supervised learning with data from multiple domains, but they do not provide transformed features for data analysis. Furthermore, EasyAdapt relies on feature concatenation; we expect to not have confounders available at inference time, which means that we cannot utilize them in the concatenated features at training time either.

Previous work which explicitly matches conditional distributions (Long *et al.*, 2013) instead uses the conditional distribution of the label given the features, rather than our approach of matching the features conditioned on the labels. It also constructs a new latent space, rather than mapping from source to target for backwards compatibility.

Optimal transport has also been proposed as a framework for domain adaptation (Courty *et al.*, 2014), and within this framework, unbalanced optimal transport has been proposed to address distribution shift (Chizat *et al.*, 2018). This relaxes the OT constraint for conservation of mass to a penalty on deviations from mass conservation, yielding improvements in label shift scenarios (Yang & Uhler, 2018). However, this approach requires challenging adversarial training (Yang & Uhler, 2018), and fails to utilize potentially available side information from confounding variables. Our work is more similar in spirit with optimal transport with subset correspondence (OT-SI) (Liu *et al.*, 2019b), which implicitly conditions on a categorical confounder (the sample's subset) to learn an optimal transport map. Our framework explicitly conditions on confounders

and is thus more general, allowing continuous, multivariate, and (using kernels) even general objects as confounding variables.

## 6 Discussion

### 6.1 Limitations

The first main limitation of our framework is that we assume access to all confounders at training time. While our experiments suggest that our approach works well given a superset of the true confounding variables, our experiments suggest that ConDo fails to benefit from receiving a subset of the true confounders. Controlling for partially-latent or fully-latent confounding remains as future work. A second limitation is that we assume a deterministic mapping (and in our concrete implementations, a linear mapping) between feature spaces. It would be nontrivial to extend our approach to non-deterministic mappings with distributional regression or conditional generative modeling. A third limitation is that, even if we are able to learn the ground-truth feature adaptation, we may only use the adapted features for downstream prediction tasks where the conditional distribution of the target variable given features is the same for source and target.

Furthermore, despite our limiting assumptions, both our proposed divergences, the reverse KLD and the MMD, suffer from non-identifiability. Multiple transformations may match the source distribution to the target distribution, and both our objective functions are indifferent among such transforms. We currently rely on gradient flow from the initial parameters to make a sensible choice, but this offers no guarantees.

### 6.2 Future work

Optimal transport (OT) offers a principled criteria, minimal transport cost, to choose among transformations which provide equal fit to the data. This suggests replacing the reverse KLD and MMD with an OT-based alternative, such as the Wasserstein distance. Yet, while minimal transport cost is an excellent "prior", it is not the only defensible choice. For example, $L_p$ regularization, empirical Bayes weight sharing such as used by ComBat (Johnson *et al.*, 2007), and constraints (e.g. non-negativity or zero-off-diagonal for linear mappings) may instead be preferred, and may be fruitfully combined with KLD and/or MMD.

Due to the affine restriction on the transformation, our proposed approach is more appropriate for adaptation settings where source and target correspond to different versions of experimental assays and similar settings where the required adaptation is affine (or even location-scale). It would be useful to examine whether our framework extends gracefully to nonlinear adaptations parameterized by neural networks.

Finally, thus far our analysis of ConDo has been purely empirical. Theoretical analysis would surely be appropriate, particularly before relying on ConDo for statistical inference tasks.

Along with our provided software, we have included code for all our experiments, in the hope that these may form a benchmark that accelerates future progress.

## 7 Conclusion

We have shown that minimizing expected divergences / distances after conditioning on confounders is a promising avenue for domain adaptation in the presence of confounded shift. Our proposed use of the reverse KLD are (to our knowledge) new in the field of domain adaptation, and may be more broadly useful. Focusing on settings where the effect of the confounder is possibly complex, yet where the source-target domains can be linearly adapted, we demonstrated the usefulness of algorithms based on our framework. These experiments show that conditioning on confounders via our ConDo framework improves the quality of learned adaptations for a variety of domains and tasks.

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

## A    Exact solution for 1d reverse KL divergence

For each $n$th sample ($1 \leq n \leq N$) drawn from the prior distribution over the confounding variable, we have obtained an estimate of its mean and variance for the source domain ($\mu_{S_n}, \sigma_{S_n}^2$) and target domain ($\mu_{T_n}, \sigma_{T_n}^2$). Then the reverse-KL objective is the following:

$$\arg\min_{m,b} \sum_{n=1}^{N} -\log(m) + \frac{m^2 \sigma_{S_n}^2}{2\sigma_{T_n}^2} + \frac{(m\mu_{S_n} + b - \mu_{T_n})^2}{2\sigma_{T_n}^2} \tag{14}$$

$$= \arg\min_{m,b} \sum_{n=1}^{N} -\log(m)(2\sigma_{T_n}^2) + m^2 \sigma_{S_n}^2 + (m\mu_{S_n} + b - \mu_{T_n})^2. \tag{15}$$

Setting the partial derivative wrt $b$ to 0, we have:

$$b^* = \frac{1}{N}[(\sum_{n=1}^{N} \mu_{T_n}) - m(\sum_{n=1}^{N} \mu_{S_n})] = \overline{\mu}_T - m\overline{\mu}_S. \tag{16}$$

Substituting this into our objective, we have

$$\arg\min_{m} \sum_{n=1}^{N} -\log(m)(2\sigma_{T_n}^2) + m^2 \sigma_{S_n}^2 + (m\mu_{S_n} + \overline{\mu}_T - m\overline{\mu}_S - \mu_{T_n})^2. \tag{17}$$

Setting the derivative wrt $m$ to 0, we obtain the following quadratic equation:

$$0 = \left[\left(\sum \sigma_{S_n}^2\right) + \left(\sum (\mu_{S_n} - \overline{\mu}_S)^2\right)\right] m^2 + \left[\sum (\overline{\mu}_T - \mu_{T_n})(\mu_{S_n} - \overline{\mu}_S)\right] m + \left[-\sum \sigma_{T_n}^2\right]. \tag{18}$$

We then apply the quadratic formula, choosing the positive solution.

## B    Implementation details and computational complexity analysis

Recall that we have $N_S$ and $N_T$ source and target samples, respectively; let us use $\tilde{N}_S$ and $\tilde{N}_T$ to denote the unique values of the source and target confounders. Let $N = \max(N_S, N_T)$ and $\tilde{N} = \max(\tilde{N}_S, \tilde{N}_T)$. We use $M = \max(M_S, M_T)$ as the feature dimension, and let the complexity of evaluating $k_{\mathcal{Z}}(z^{(n_1)}, z^{(n_2)})$ be $O(C)$. Recall that $K_{\mathcal{X}}$ is the number of multiple imputations per $Z$ value. Also recall that $K_{\mathcal{Z}}$ is the number of sampled $Z$ values used per ConDo-MMD minibatch.

The complexity of computing the product prior weights is $O(C\tilde{N}^2)$. We achieve this speedup by, rather than assigning weights to each sample, instead assigning weights to each unique value of the confounder and multiplying the weights from Eq. (9) with the number of observations of each unique value.

When confounding variables are discrete, conditional sampling has negligible runtime cost. When confounders are continuous, the runtime is dominated by MICE-Forest (Wilson *et al.*, 2022) training LightGBM (Ke *et al.*, 2017) models for each feature given all the other features. The runtime of each LightGBM model is $O(NM)$, we run the MICE imputation algorithm for a fixed number (2) of iterations, and generate $K_{\mathcal{X}}$ imputations, so the total runtime complexity for this step is $O(NM^2 K_{\mathcal{X}})$.

The space complexity of conditional sampling is potentially burdensome. Compared to input datasets of size $N_S \times M_S$ and $N_T \times M_T$, we generate datasets of size $K_{\mathcal{X}} N \times M_S$ and $K_{\mathcal{X}} N_S \times M_S$, for source and target, respectively. In our provided software, we implement an option to subsample a fraction $F \leq 1$ of $N$ samples at each epoch before performing conditional sampling, reducing datasets to size $FK_{\mathcal{X}} N \times M_S$ and $FK_{\mathcal{X}} N_S \times M_S$.

For ConDo-KLD, we compute means and (inverse) covariances for each confounder value. Computing these statistics from the conditionally sampled data, for continuous confounders, has complexity $O(\tilde{N} K_{\mathcal{X}} M^2 + \tilde{N} M^3)$.

Computing these from raw data, for discrete confounders, has complexity $O(NM^2 + \tilde{N}M^3)$. Computing the ConDo-KLD objective at each Newton conjugate gradient iteration has complexity $O(\tilde{N}M^3)$.

Evaluating ConDo-MMD on a single minibatch has complexity $O(K_{\mathcal{Z}}K_{\mathcal{X}}^2 M)$; this is substantially less than that of full-batch optimization, which would be $O(\tilde{N}N^2M)$.

## C  Additional Material for the Experiments

### C.1  Synthetic 1d feature with 1d continuous confounder

In the source domain, the confounder $Z_S^{(n)} \sim \text{Uniform}[4, 8]$; in the target domain the confounder $Z_T^{(n)} \sim \text{Uniform}[0, 8]$. For the homoscedastic scenario, $X^{(n)} \sim \mathcal{N}\left(4Z^{(n)} + 1, (1)^2\right)$. For the heteroscedastic scenario, $X^{(n)} \sim \mathcal{N}\left(4Z^{(n)} + 1, (Z^{(n)} + 1)^2\right)$. For the nonlinear scenario, $X^{(n)} \sim \mathcal{N}\left(4[\text{ReLU}(Z^{(n)} - 5)]^2 + 1, ([\text{ReLU}(Z^{(n)} - 5)]^2 + 1)^2\right)$. Then, for the source domain, we have $X_S^{(n)} = 2X^{(n)} + 5$; for the target domain, we have $X_T^{(n)} = X^{(n)}$.

In all these experiments, we used the default hyperparameters, with the exception of using learning rate 0.01 for 100 epochs, for both MMD and ConDo-MMD.

In Figure S1, we show the error in estimation of the feature alignment parameters ($\boldsymbol{A}, \boldsymbol{b}$). ConDo outperforms the baselines when target shift is present, and is not systematically worse when no shift or only feature shift is present.

We show train errors in Figure S2, with-and-without label shift, with-and-without feature shift, and with-and-without additional noise, for a total of 8 settings.

### C.2  Synthetic 1d feature with multi-dimensional continuous confounders

In all these experiments, we used the default hyperparameters, with the exception of using learning rate 0.01 for 100 epochs, for both MMD and ConDo-MMD.

In Figure S3, we show the rMSE on the training set for multi-dimension contiguous confounder experiments.

In Figure S4, we show the effect of running ConDo with partially-observed confounding variables. We see that, on the one hand, ConDo is typically non-inferior to the baselines when confounders are partially observed. On the other hand, ConDo performance quickly degrades as the number of observed confounders is reduced, so that when half of the confounding variables are observed, ConDo is not better than the baselines.

### C.3  Synthetic 1d and 2d features with 1d categorical confounder

In all these experiments, we used the default hyperparameters, with the exception of using learning rate 0.01 for 100 epochs, for both MMD and ConDo-MMD.

#### C.3.1  Synthetic 1d feature with 1d categorical confounder

Figure S5 provides additional results for our experiments with a 1d feature confounded by a 1d categorical confounding variable. We see that both ConDo methods match the true distribution better than the baseline methods. In particular, as sample size increases, ConDo MMD converges towards the true target distribution. We confirm this in Figure S6, showing that ConDo leads to lower-error estimates of the true mapping, and that ConDo MMD in particular has decreasing error as a function of sample size.

#### C.3.2  Synthetic 2d features with 1d categorical confounder

The procedure for generating data in this scenario is adapted from the Python Optimal Transport library (Flamary *et al.*, 2021). The batch-effected source data are generated first, with $N = 200$ split between two

circles centered at $(0,0)$ and $(0,2)$; the points are distributed with angle distributed iid around the circle from $U[0, 2\pi]$, and with radius sampled iid from $\mathcal{N}(0,1)$.

Next, we analyze the performance of our approach on 2d features requiring an affine transformation. We also use this setting to assess the downstream performance of classifiers which are fed the adapted source-to-target features. The synthetic 2d features, before the batch effect, form a slanted "8" shape, shown in blue/green in Figure S7. A linear classifier separating the upper and lower loops is depicted in cyan, for both the source and target domains.

Our results are shown in Figure S7. On the left column, we compare methods in the case where there is no confounded shift. (This setting is from Python Optimal Transport (Flamary *et al.*, 2021).) In the middle column, we have induced a confounded shift: One-fourth of the source domain samples come from the upper loop of the "8", while half of the target domain samples come from the upper loop. This allows us to assess the affects of confounded shift on downstream prediction of the confounder (up-vs-down), as well as a non-confounder (left-vs-right). In the right column, we have induced a confounded shift as before, while making the true source-target transform more challenging, by using a randomly-generated true affine transform. The affine transform matrix is set to be random yet positive-determinant via

$$\boldsymbol{A} = \begin{bmatrix} \cos\phi_1 & -\sin\phi_1 \\ \sin\phi_1 & \cos\phi_1 \end{bmatrix} \begin{bmatrix} d_1 \\ d_2 \end{bmatrix} \begin{bmatrix} \cos\phi_2 & -\sin\phi_2 \\ \sin\phi_2 & \cos\phi_2 \end{bmatrix},$$
$$\phi_1, \phi_2 \sim \text{Unif}[-\pi, \pi],$$
$$d_1, d_2 \sim \text{Unif}[0.5, 1.5].$$

As shown in Figure S7, all methods perform similarly where there is no confounded shift (left column), but the vanilla domain adaptation approaches fail in the presence of confounding (center and right columns).

In Figure S8, we show performance on estimating the feature mapping parameters. ConDo outperforms the corresponding baselines.

## C.4 ANSUR II anthropometric survey data

In all ANSUR II experiments, we used the hyperparameters given in the main text. We used the default parameters for TabPFN. In Figure S9, we show results for the same experimental setup as Figure 6, except that TabPFN prediction models are instead trained on adapted source-to-target features, then applied on target features. We see the same pattern, with ConDo methods outperforming their baseline counterparts, especially for the affine transformation scenario.

In Figure S10, we show feature-space mapping parameter estimation performance. In this setting, MMD, ConDo Gaussian KLD, and ConDo MMD perform equally well. Meanwhile, Gaussian OT performs poorly on the affine transform scenario, while outcompeting the others on the location-scale transform scenario.

In Figure S12, we show results for target feature data simulated with no feature shift ($\boldsymbol{A} = I_d, \boldsymbol{b} = \boldsymbol{0}$), but with a male-female split of 75%-25% in source and 25%-75% in target. As before, we show domain adaptation methods attempting to learn affine and location-scale feature transformations. In Figure S11, we show results for target feature data simulated with no shift in the confounder distribution (probability of sampling male and female each set to 0.5 for both source and target), but with domain adaptation methods attempting to learn affine and location-scale feature transformations. In Figure S13, we show results for target feature data simulated with no change in the shift in the confounder distribution and with no feature shift, but with domain adaptation methods attempting to learn affine and location-scale feature transformations.

## C.5 Image color adaptation

The day photo is $669 \times 1000$, leading to a dataset of size $669{,}000 \times 3$ (for 3 RGB channels). The sunset photo is $750 \times 1000$, while the beach photo is $723 \times 964$.

Both MMD and ConDo MMD are run for 100 epochs with the AdamW optimizer (Loshchilov & Hutter, 2017) with learning rate of $10^{-2}$ and weight decay of $10^{-4}$. We use a batch size of 8; each "sample" in the batch

consists of the MMD loss over paired 20-pixel datasets. Each epoch is defined as a single pass-through over randomly generated $0.01 * \min(669 \times 1000, 750 \times 1000)$ unique paired 20-pixel datasets (for the day-sunset task, and analogously for the other task). This randomly-generated dataset of datasets is reused across epochs.

In Figure S14 we see that the ConDo methods do better on Sunset $\longrightarrow$ Beach, producing a brighter image, particularly for the clouds and the horizon. We see that both MMD and ConDo MMD fail to produce good inverse mappings.

### C.6  California housing price prediction

The California Housing dataset has 20,640 samples (before subsampling as described in the main text), and the classifier receives 7 input features (with median income removed). In all experiments, we used the hyperparameters given in the main text. We used the default parameters for the TabPFN classifier.

In Figure S15, we repeat the previous experiment, except that TabPFN prediction models are instead trained on adapted source-to-target features on the train-set; then the model is provided target test-set features directly. We see the same pattern as before, with ConDo methods outperforming their baseline counterparts.

### C.7  SNAREseq single-cell multi-omics dataset

For our experiment with SNAREseq data (Demetci *et al.*, 2022), the ATAC-seq dataset has 19 features, and the RNA-seq dataset has 10 features. The data was preprocessed in (Demetci *et al.*, 2022), including unit normalization as in (Demetci *et al.*, 2022). In all SNAREseq experiments, we used the hyperparameters given in the main text.

In Figure S16, we show results for the setting where classifiers were instead trained on adapted source-to-target features. In other words, using the paired samples, we learned a transformation from source domain to target domain. We then applied this transformation to the 500 source domain training samples, and fitted the classifier on these transformed features. Finally, we applied the classifier to (untransformed) target domain features, and then evaluated predictions.

### C.8  Gene expression microarray batch effect correction

In all bladderbatch experiments, we used the hyperparameters given in the main text, with the exception of using learning rate 0.01 for 100 epochs, for both MMD and ConDo-MMD.

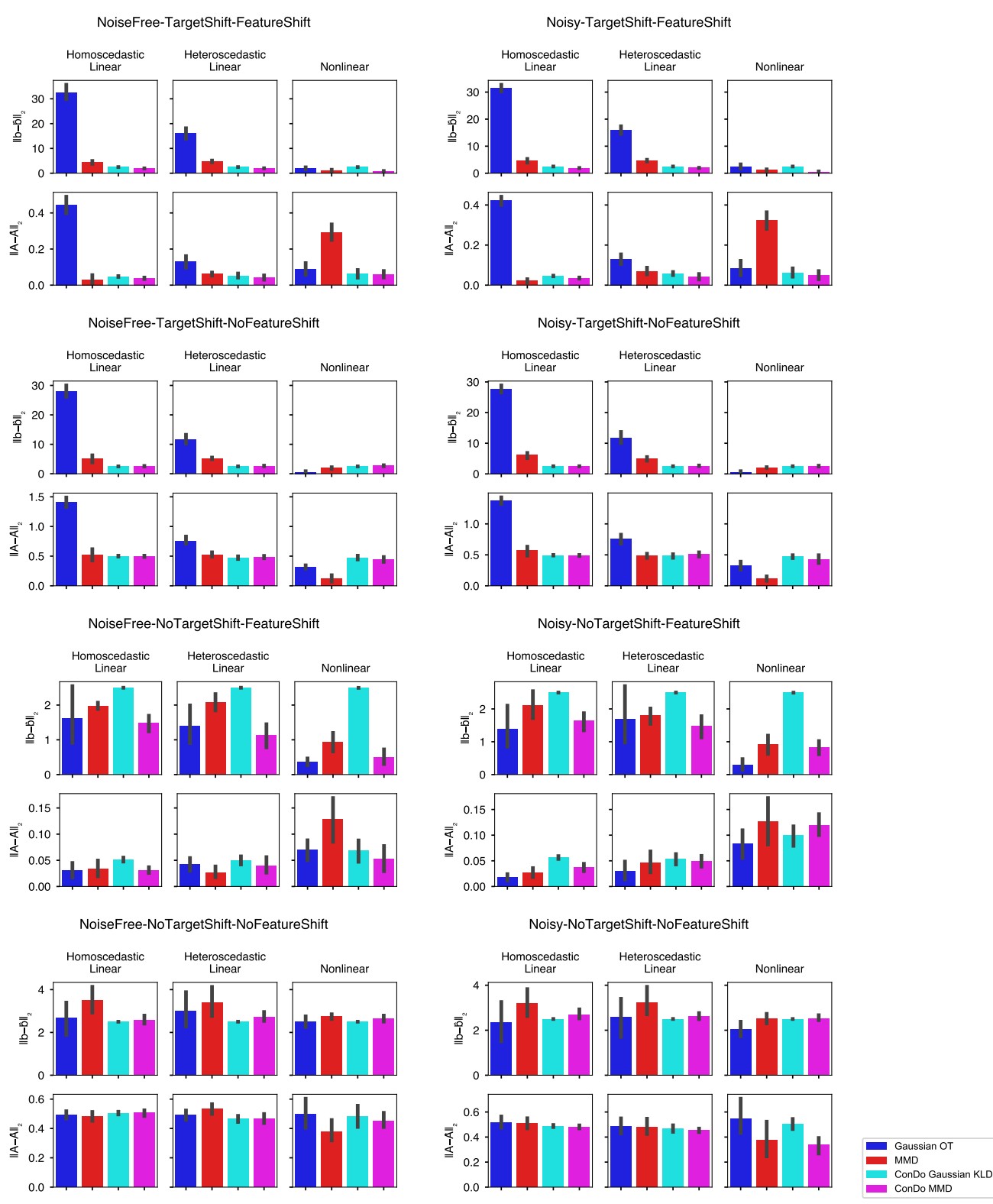

Figure S1: Parameter estimation for transform of 1d data with a continuous confounder. We depict the error in estimating $\boldsymbol{b}$ and $\boldsymbol{A}$, respectively, with the $L_2$ vector norm and induced $L_2$ matrix norm (spectral norm). The error bars in the barplots depict standard deviations over the 10 random simulations.

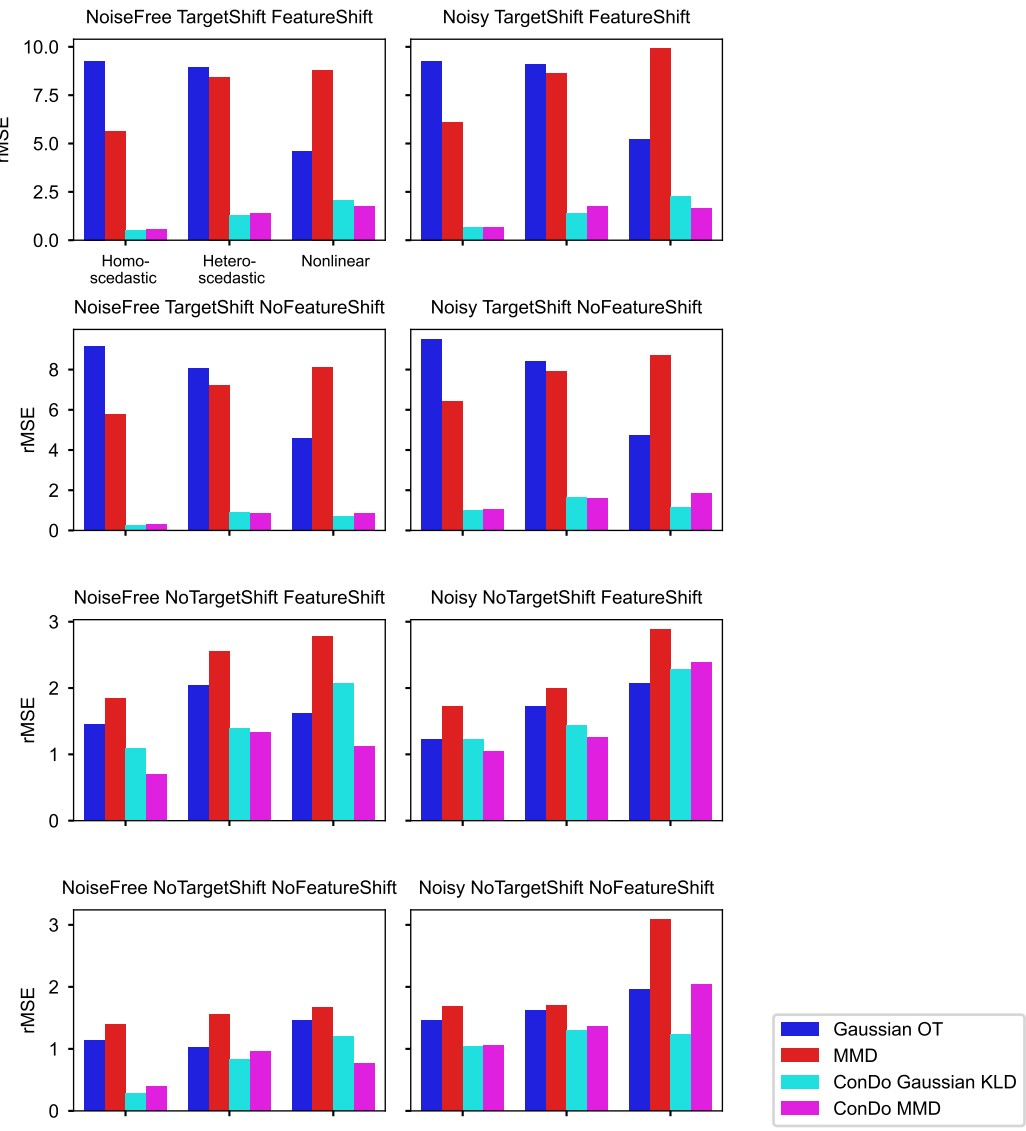

Figure S2: Train errors for experiment with synthetic 1d features with 1d continuous confounder. For each adaptation method, we compute the rMSE of true target feature values vs inferred target feature values after adaptation, averaged over 10 simulations.

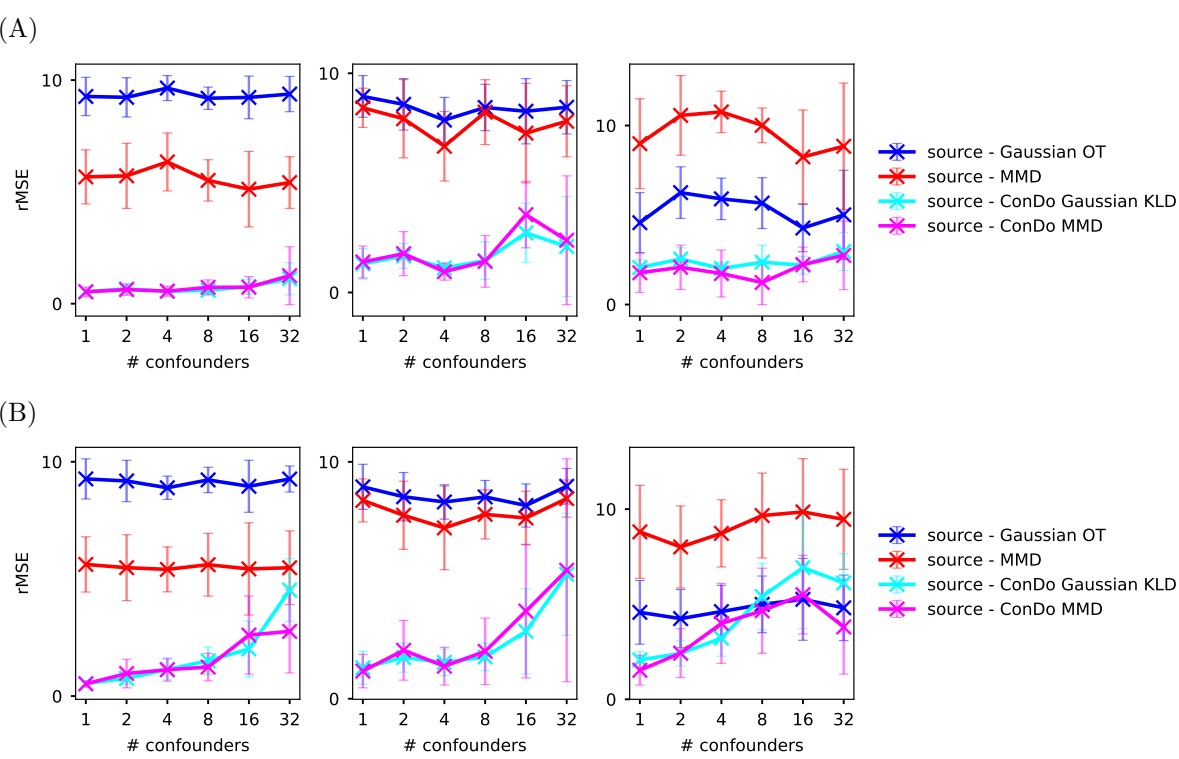

Figure S3: Results on training data for transforming 1d data with multiple continuous confounders, with extra irrelevant $\mathcal{N}(0, 1)$ confounders, shown in (A), and with noisy additive decomposition, shown in (B). The rMSEs are averaged over 10 random simulations are shown for training data (200 samples per simulation). The columns, in order, correspond to a confounder with a linear homoscedastic effect, a confounder with a linear heteroscedastic effect, and a confounder with a nonlinear heteroscedastic effect.

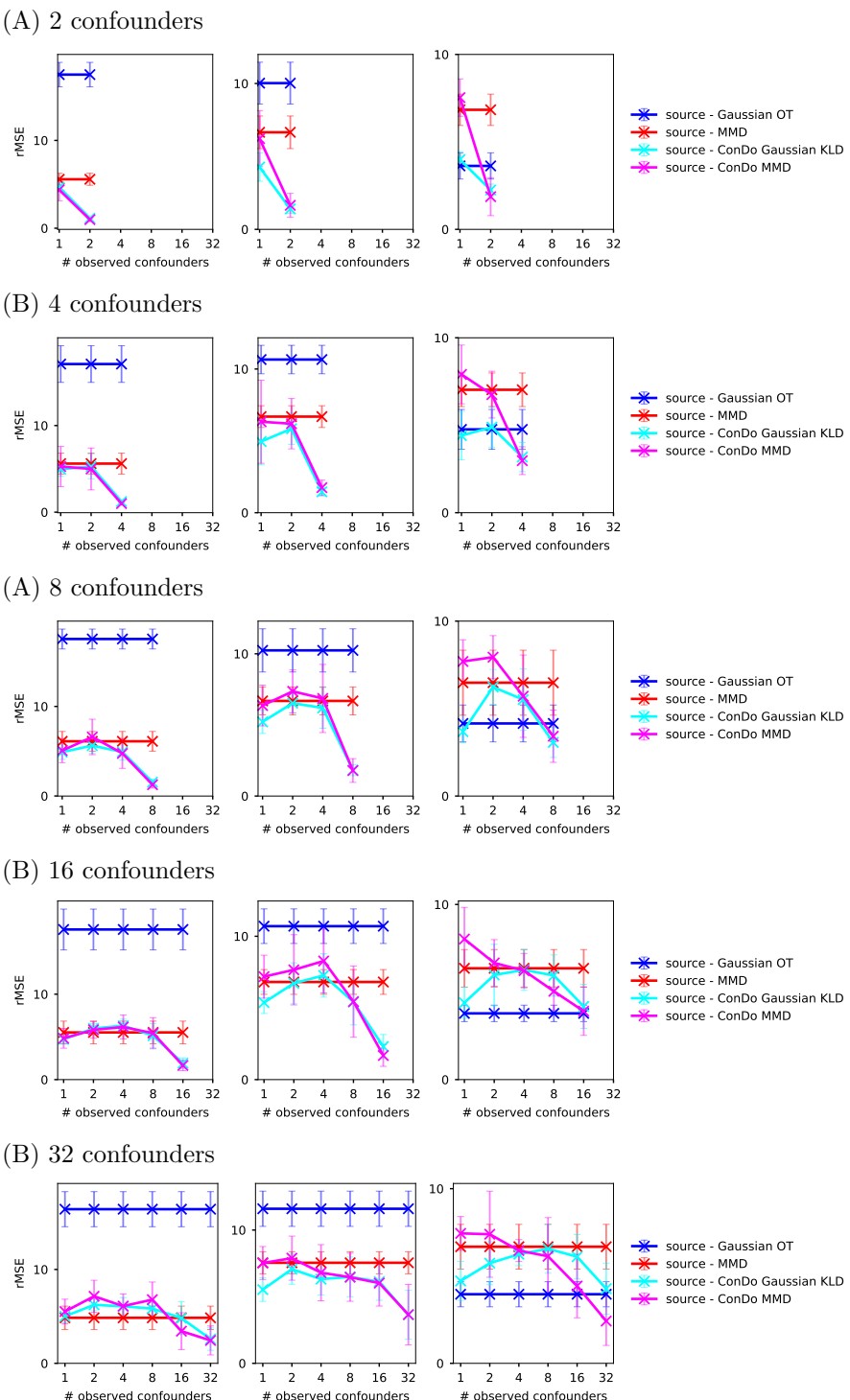

Figure S4: Results for transforming 1d data with multiple continuous confounders with noisy additive decomposition, while varying both the number of actual confounders and the number provided to ConDo methods. In each row, we show results for a given number of actual confounders; within each plot, we show ConDo performance as a function of how many were provided to ConDo. The rMSEs are averaged over 10 random simulations are shown for heldout test data (100 samples per simulation). The columns, in order, correspond to a confounder with a linear homoscedastic effect, a confounder with a linear heteroscedastic effect, and a confounder with a nonlinear heteroscedastic effect.

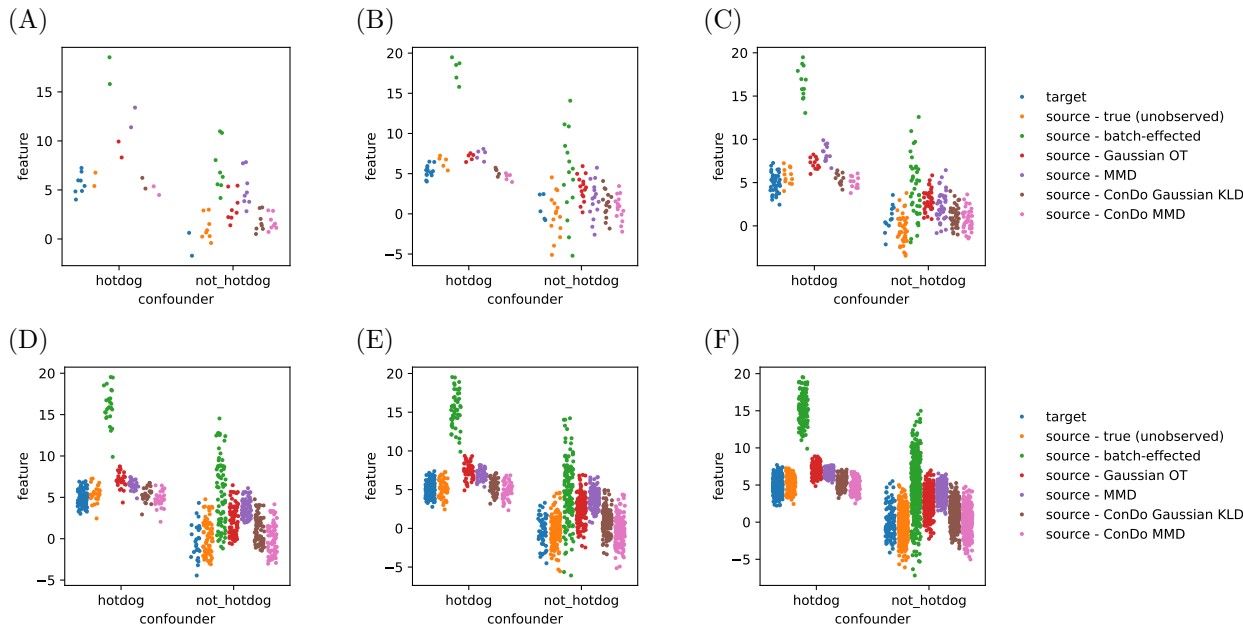

Figure S5: Additional results for 1d feature confounded by 1d category. Results are shown (A) $N = 10$, (B) $N = 20$, (C) $N = 50$, and (D) $N = 100$, (E) $N = 200$, and (F) $N = 500$ samples in each of the source and target training sets. We depict the original, latent, shifted, and adapted data, for each value of the categorical confounder.

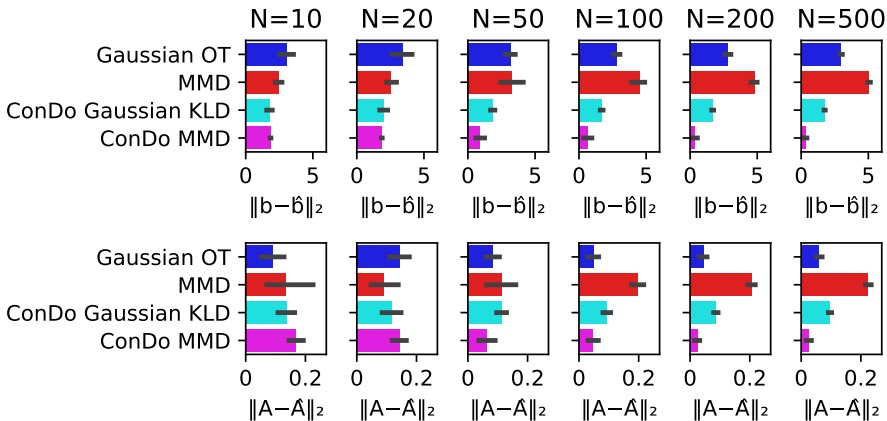

Figure S6: Parameter estimation performance for transform of 1d data with a categorical confounder. We depict the error in estimating $\boldsymbol{b}$ and $\boldsymbol{A}$, respectively, with the $L_2$ vector norm and induced $L_2$ matrix norm (spectral norm). The error bars in the barplots depict standard deviations over the 10 random simulations. Columns correspond to varying sample sizes.

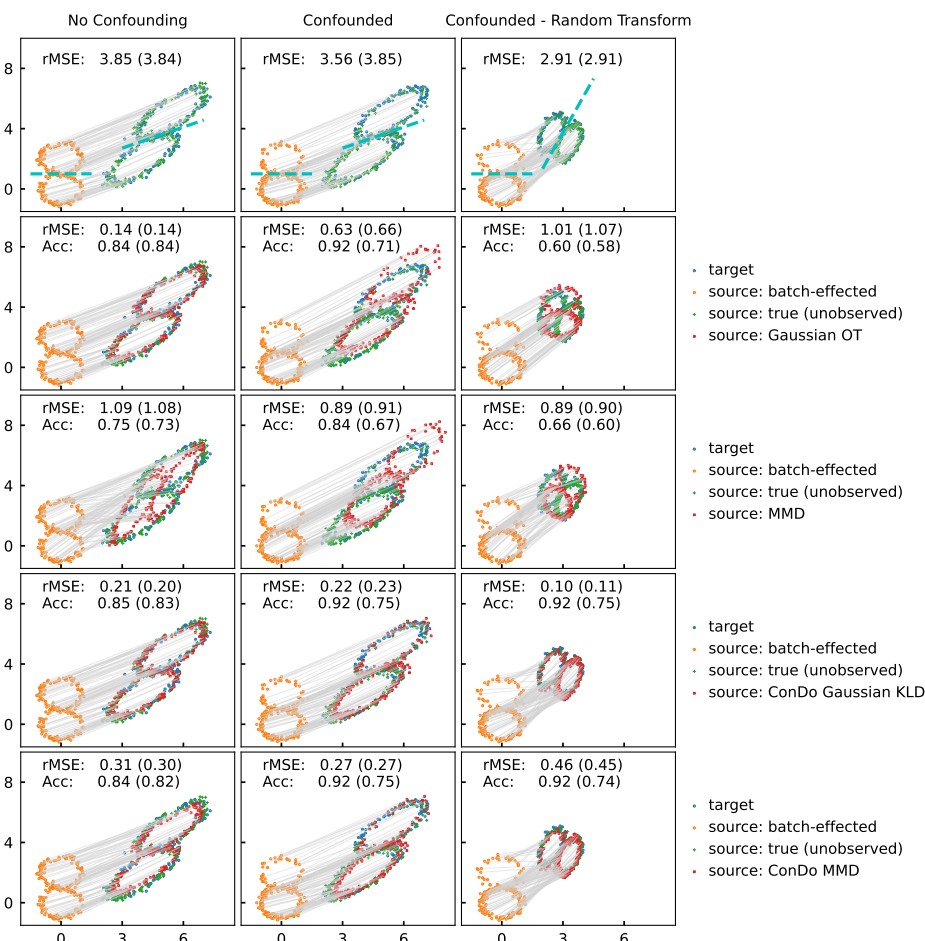

Figure S7: Results of affine transform of 2d data with a categorical confounder. We print the rMSE and accuracy on the training data (and on heldout test data in parentheses). These values are the result of averaging over 10 random simulations, while the plot is generated from the final simulation.

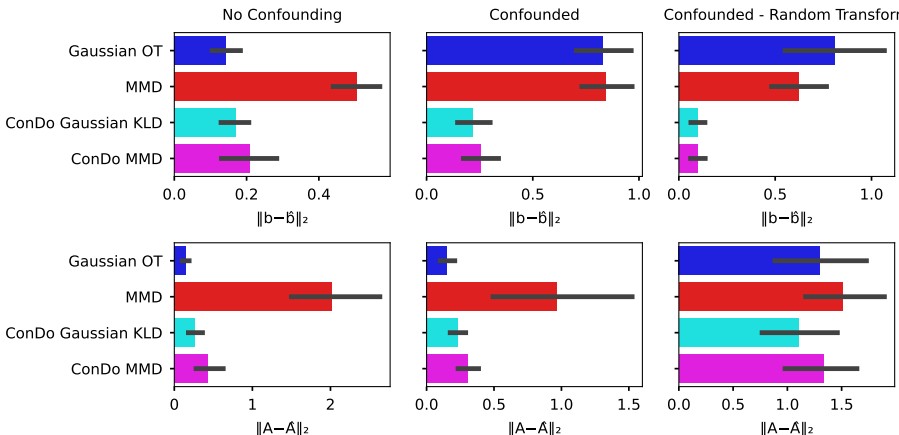

Figure S8: Parameter estimation for affine transform of 2d data with a categorical confounder. We depict the error in estimating $b$ and $A$, respectively, with the $L_2$ vector norm and induced $L_2$ matrix norm (spectral norm). The error bars in the barplots depict standard deviations over the 10 random simulations.

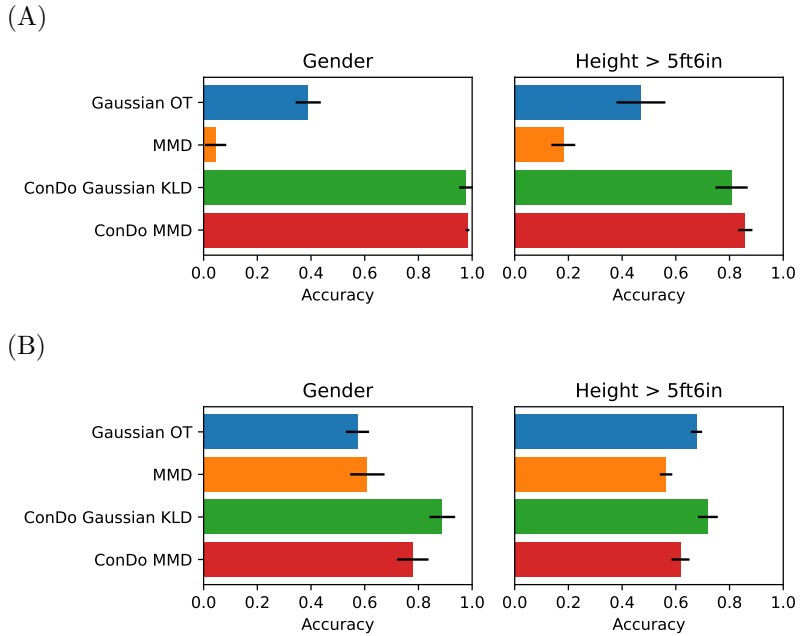

Figure S9: Results on ANSUR II, for the setting where TabPFN prediction models are trained on adapted source-to-target features and applied to target features, for affine transformation (A) and location-scale transformation (B).

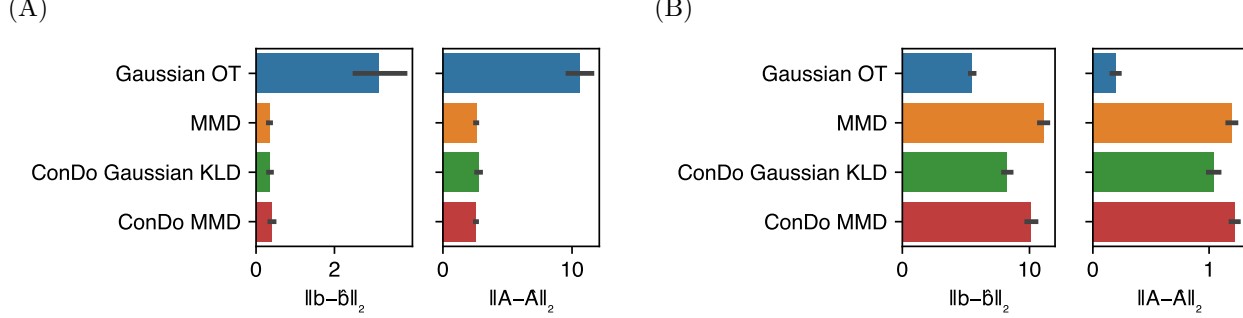

Figure S10: Feature mapping parameter estimation performance on ANSUR II, for affine (A) and location-scale (B) transforms. We depict the error in estimating $b$ and $A$, respectively, with the $L_2$ vector norm and induced $L_2$ matrix norm (spectral norm). The error bars in the barplots depict standard deviations over the 10 random simulations.

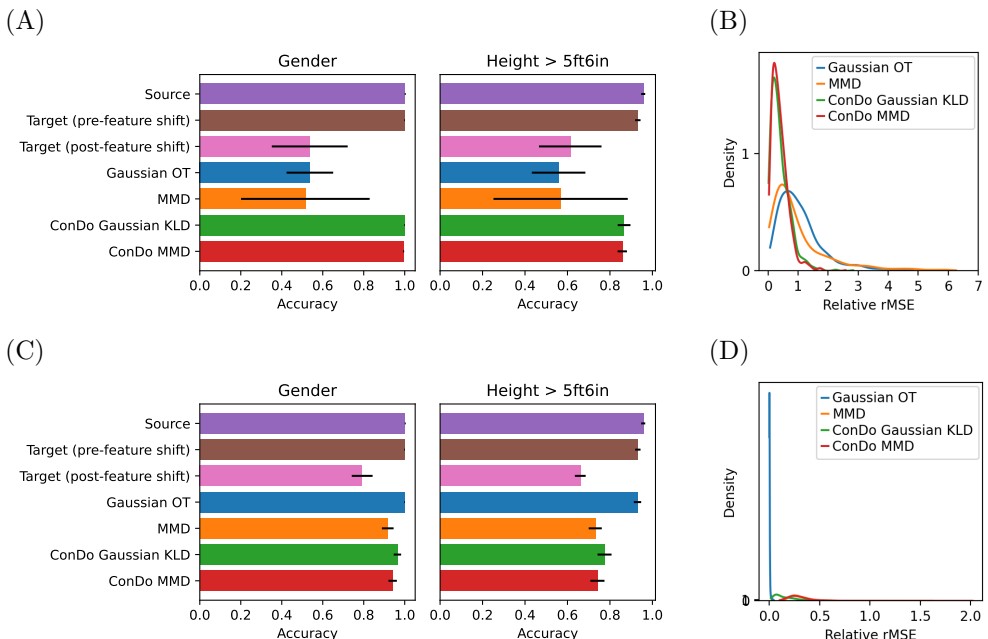

Figure S11: Results on ANSUR II, where the data is generated with only feature shift, for affine transformation (A-B) and location-scale transformation (C-D).

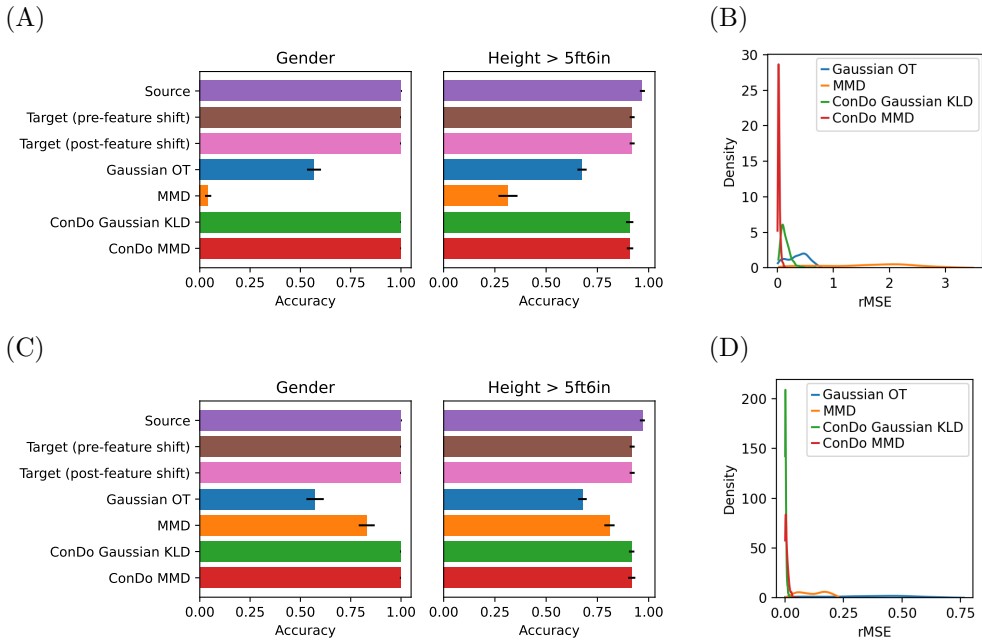

Figure S12: Results on ANSUR II, where the data is generated with no feature shift but shift in the confounding gender distribution, for affine transformation (A-B) and location-scale transformation (C-D). Relative rMSE is not available, because pre-adaptation rMSEs are all zero, so raw rMSE is depicted instead.

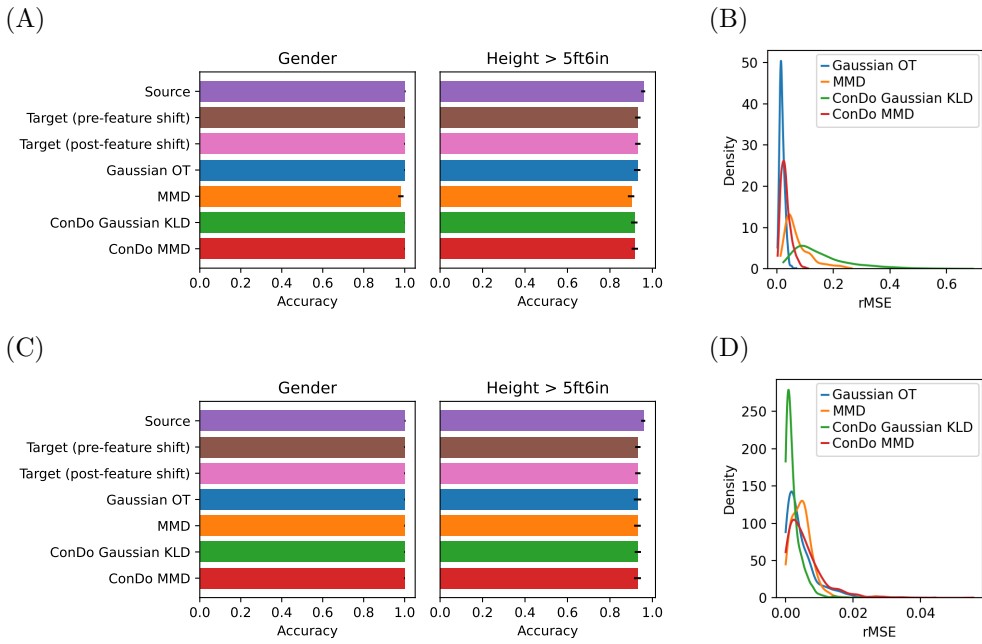

Figure S13: Results on ANSUR II, where no shift at all takes place, for affine transformation (A-B) and location-scale transformation (C-D).

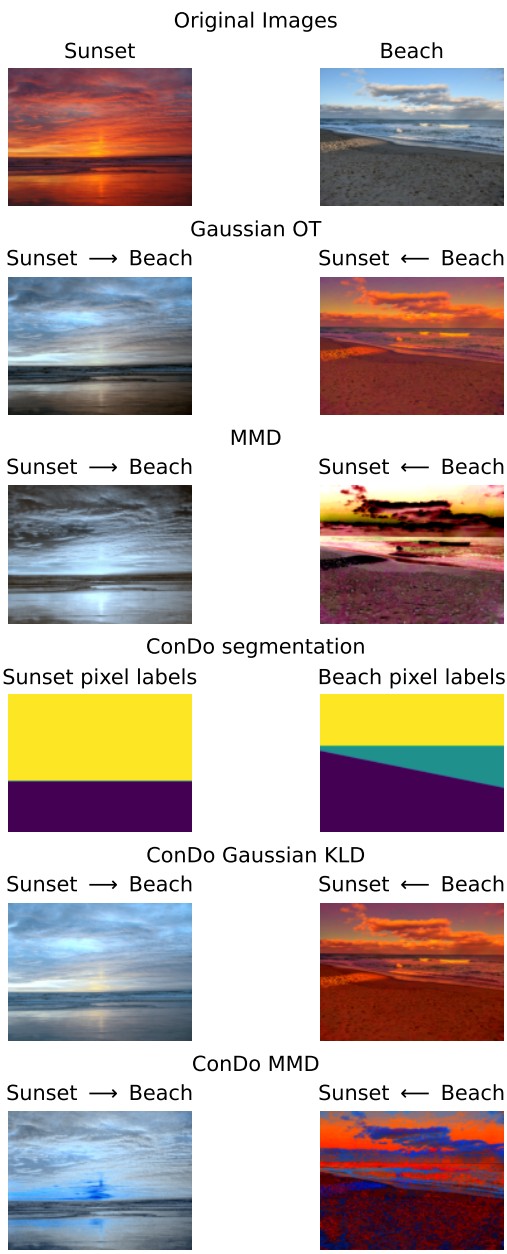

Figure S14: Image color adaptation results with confounded shift between Sunset and Beach image.

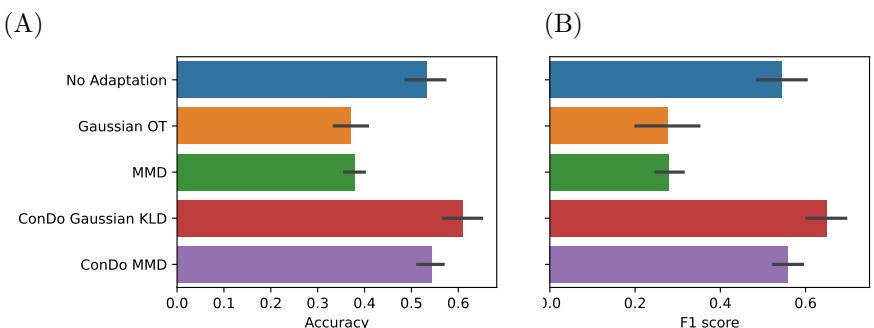

Figure S15: Results on the California Housing dataset, for the setting where TabPFN prediction models are trained on adapted source-to-target features and applied to target features. We show accuracy (A) and F1-score (B) on target test data, over 10 random simulations, with error bars depicting the standard deviation over simulations.

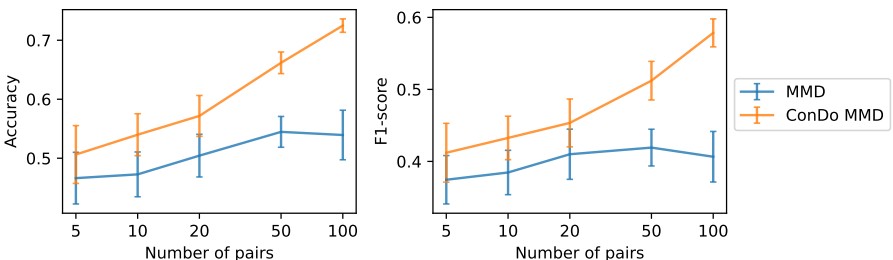

Figure S16: SNAREseq cell type classifier performance on test set. Classifier performance (accuracy and F1-score) is shown as a function of the number of paired samples available to the domain adaptation methods. The error bars depict the standard error computed over 10 random simulations.

