# OpenReview forum: "Towards Backwards-Compatible Data with Confounded Domain Adaptation"
_TMLR — Accepted by TMLR_

### Review · Reviewer_bEdS · 2024-07-15

**Summary Of Contributions:**

This paper presents a new framework for domain adaptation that addresses simultaneous covariate and label shifts, confounded with each other, by modifying generalized label shift (GLS). The method ensures backward compatibility, allowing adapted data to be used in existing models without re-training. Implemented using Gaussian reverse Kullback-Leibler divergence and maximum mean discrepancy, the framework is validated on synthetic and real datasets, demonstrating its effectiveness and significant improvement over traditional methods.

**Audience:**

Yes

**Claims And Evidence:**

Yes

**Requested Changes:**

Please see the comments of weaknesses.

**Strengths And Weaknesses:**

Strengths:
1. The paper's presentation is clear. It also provides detailed explanations from a theoretical perspective.
2. The proposed method mostly involves modifying the loss. The training procedure is clean and easy to implement.
3. The problem the paper focuses on is novel, i.e., Confounded Domain Adaptation.

Weaknesses:
1. Figure 1 is misleading. Detailed illustrations are expected to well understand the figure.
2. The current implementation assumes a deterministic and linear mapping between feature spaces, which may not be suitable for all types of data or adaptation tasks. Extending this to non-deterministic or nonlinear mappings would enhance the framework's versatility.

---

> ### Author Response · Authors · 2024-09-16
> **Re: Review of Paper2911 by Reviewer bEdS**
>
> Thank you for your time and valuable feedback.
>
> > "Figure 1 is misleading. Detailed illustrations are expected to well understand the figure."
>
> Thank you for this feedback. We have added a full explanation in the figure's caption, and welcome further feedback.
>
> > "The current implementation assumes a deterministic and linear mapping"
>
> Please see shared response.

---

### Review · Reviewer_LMYA · 2024-08-27

**Summary Of Contributions:**

This paper introduces a novel framework called ConDo (Confounded Domain adaptation) for performing domain adaptation in the presence of confounded shift between source and target domains. The key idea is to minimize the expected divergence between conditional distributions of source and target features, conditioning on confounding variables. The authors propose concrete implementations using Gaussian reverse KL divergence and maximum mean discrepancy (MMD) as divergence measures. They evaluate ConDo on synthetic and real datasets, showing improvements over baseline domain adaptation methods, especially in scenarios with both covariate shift and label shift.

**Audience:**

Yes

**Claims And Evidence:**

Yes

**Requested Changes:**

- Explore scenarios with partial observability of confounders, or methods to infer potential confounders when not all are known. You could see this happen more in text domains given the high dimensional data.
- Develop theoretical guarantees or error bounds for the proposed methods, even if under simplifying assumptions. This would strengthen the theoretical contributions.

**Strengths And Weaknesses:**

## Strengths
- The paper addresses an important and practical problem in domain adaptation---handling scenarios where both covariate shift and label shift occur simultaneously and are confounded.
- The proposed ConDo framework is general and flexible, allowing for different divergence measures and types of confounders (continuous, categorical, etc.).
- The theoretical formulation is sound, building on and extending concepts from optimal transport and generalized label shift.
Extensive empirical evaluation on synthetic and real datasets demonstrates the effectiveness of the approach, especially in challenging confounded shift scenarios.
- The authors provide a thoughtful analysis of when and why their method outperforms baselines, as well as discussing limitations. The paper is well-written and structured, with clear explanations of concepts and methods.

## Weaknesses
- Unless I missed something, the paper assumes access to all confounders at training time, which may be unrealistic in some real-world scenarios. More discussion on partial observability of confounders would be valuable.
- The current implementation is limited to linear/affine transformations between domains. While this is justified for certain applications, extending to nonlinear transformations would broaden the applicability.
- While the empirical results are promising, including those in the supplementaries, the paper lacks theoretical guarantees or error bounds for the proposed methods. There's nothing wrong with a fully empirical TMLR paper but having these would certainly strengthen the paper.

---

> ### Author Response · Authors · 2024-09-16
> **Re: Review of Paper2911 by Reviewer LMYA**
>
> Thank you for your time and valuable feedback.
>
> > "The current implementation is limited to linear/affine transformations between domains."
>
> Please see main response.
>
> > "the paper lacks theoretical guarantees"
>
> Please see main response.
>
> > "More discussion on partial observability of confounders would be valuable."
>
> Thank you for this idea. We performed additional experiments, using the setting in Section 4.2, where the number of actual confounders varied in {1, 2, 4, 8, 16, 32}. For each given number of actual confounders, we varied the number of confounders provided to ConDo  -- eg for 16, we varied the number provided in {1, 2, 4, 8, 16}. See results below. The results substantiate our claim that ConDo can utilize a superset but not a subset of the true confounders. We believe that addressing this challenge is best left to future work; please see main response.
>
> In Section 4.2 and C.2, we added:
> "We furthermore use this latter setting to examine the behavior of ConDo when confounders are partially observed. Results, shown in Figure S4, indicate that typically ConDo fails to take advantage of partially-observed confounders while remaining non-inferior to baselines. ... In Figure S4, we show the effect of running ConDo with partially-observed confounding variables. We see that, on the one hand, ConDo is typically non-inferior to the baselines when confounders are partially observed. On the other hand, ConDo performance quickly degrades as the number of observed confounders is reduced, so that when half of the confounding variables are observed, ConDo is not better than the baselines."
>
> In Limitations, we changed:
> "While our experiments suggest that our approach works well given a superset of the true confounding variables, we do not expect it to perform well with only a subset of the true confounders." -> "While our experiments suggest that our approach works well given a superset of the true confounding variables, our experiments suggest that ConDo fails to benefit from receiving a subset of the true confounders. Controlling for partially-latent or fully-latent confounding remains as future work."

---

> > ### Comment · Reviewer_LMYA · 2024-10-08
> > **Thank you**
> >
> > I appreciate your thoughtful comments and the additional experiments. They're very helpful.

---

### Review · Reviewer_Umaf · 2024-09-13

**Summary Of Contributions:**

I have reviewed this paper when it was first submitted several years ago so I will skip directly to the comments related to discussing what has changed since then.

**Audience:**

Yes

**Claims And Evidence:**

Yes

**Requested Changes:**

1. A suggestion from my side would be to indicate for all experiments what is V1 data, what is V2 training data, and what is V2 test data accompanied by the confounders between the training and test sets.

2. For all tested synthetic cases, can we compare the found affine mapping with the true one as requested by the editor in the previous round of reviews?

3. Give an example of a commonly used domain adaptation benchmark where the confounding shift doesn't need to be defined artificially.

**Strengths And Weaknesses:**

After reviewing this paper for the second time in two years, I still find it very hard to parse. I will skip the technical details of this work in my review as they didn’t change much and they were already solid and well-justified at the time of the first submission. I will mainly concentrate on the positioning with respect to the related works and the experimental results that have changed since the last submission. (I believe that evaluating the scikit-learn-friendly implementation mentioned by the authors is out of scope of this review).
Regarding the two criteria mentioned above, I note the following:

1. Confounded shift vs. GTS: the only difference here, as stated by the authors, is that GTS will correct the shift between the conditional distributions given the target variable, while the authors would like to have a more general setup where the target variable may change at inference time and is not observed. My question then is as follows: is it a meaningful setup given that the target variable (confounder) at test time may be arbitrary and no assumption is made about it? How identifiable is such a setting? I would expect that generally speaking there is no solid statistical reason to believe one can find a mapping that will be invariant for some unobserved target variables if no assumptions are made about the latter.

2. The experiments (new and old) are not helpful in better illustrating the need for such a setup as almost all of them (please correct me if I’m wrong) are artificial in how the confounded shift is defined. Is there any natural application of this idea where the shift is real-world and the confounders change from V2 training to V2 inference? The only experiment I found useful in this sense is the one related to ANSUR II data. This setup echoes the one presented in Figure 1 with a confounder that changes between V2 training (male vs. female) and test (height). In this sense, the obtained results seem intuitive as the two confounders are pretty correlated. I guess some hypothesis of this kind is needed to make this setup well-posed as well.

3. The assigned editor mentioned in their last review that it would be great to compare the learning mapping to the true ground truth mapping behind the shift. Is it possible for the authors to add this experiment?

---

> ### Author Response · Authors · 2024-09-16
> **Re: Review of Paper2911 by Reviewer Umaf**
>
> Thank you for your time and valuable feedback.
>
> > Confounded shift vs. GTS: the only difference here, as stated by the authors, is that GTS will correct the shift between the conditional distributions given the target variable, while the authors would like to have a more general setup where the target variable may change at inference time and is not observed.
>
> It's a bit subtle, but what is new about "Confounded Domain Adaptation" is more than what is new about "Confounded Shift vs GTS". What is new about Confounded Shift is the asymmetry such that we must not map both domains to a new latent space, and this is why its "extended representation" includes the domain info. What is new about Confounded Domain Adaptation is that, in addition, we constrain g(X) to be constant in terms of Z because Z is not always observed at test time.
>
> > My question then is as follows: is it a meaningful setup given that the target variable (confounder) at test time may be arbitrary and no assumption is made about it? How identifiable is such a setting?
>
> We added the following to the main text: "A third limitation is that, even if we are able to learn the ground-truth feature adaptation, we may only use the adapted features for a downstream prediction task if the conditional distribution of that downstream variable given features is the same for source and target." As noted in the paper, identifiability is a problem for inferring the true V2-features to V1-features adaptation. But given an adaptation that allows us to infer the true V1-features and a still-correct previously-learned model for a downstream target given V1-features, one can simply compose the two functions to obtain a prediction model for that downstream task.
>
> > Is there any natural application of this idea where the shift is real-world...
>
> As discussed in the Introduction and its references (Hicks et al., 2018; Zindler et al., 2020; Antonsson & Melsted, 2024), this type of shift is close-to-universal in biology. There will be statistical differences in the logistics of collecting data from healthy vs sick patients, statistical differences in protocols in a time course experiment on lab mice even as the surviving mouse population changes in composition over time, etc.
>
> > ... the confounders change from V2 training to V2 inference? The only experiment I found useful in this sense is the one related to ANSUR II data.
>
> (We may be misunderstanding this question.) We do not assume any change in distribution between V2 training and V2 inference. The conditional distribution of Height | V2-features is the same between V2 training and V2 test; and the conditional distribution of MaleVsFemale | V2-features is the same between V2 training and V2 test. We also do not assume that the confounding variable "changes" from MaleVsFemale to Height; predicting Height | V2-features is simply an example of a downstream task enabled by correctly inferring the V2-to-V1 mapping.
>
> What we do assume is that the confounding variable (MaleVsFemale) is available at V2 training and that this is sufficient to learn a mapping from V2-features to V1-features. And then, to predict Height | V2-features, we simply compose the feature alignment g mapping V2-features to V1-features with the regression model Height | V1-features (which continues to be valid, because the condition distribution did not change), since Height | V1-features = Height | g(V2-features).
>
> > In this sense, the obtained results seem intuitive as the two confounders are pretty correlated.
>
> It is true that in this case, the confounder MaleVsFemale and the downstream task target Height are correlated. But such correlation isn't strictly necessary. What is necessary is that we correctly map from V2-to-V1; then any downstream prediction model that uses V1 can be reused, as long as the downstream conditional distribution has not changed.
>
> > A suggestion from my side would be to indicate for all experiments what is V1 data...
>
> Thank you for this suggestion, which we have incorporated in the latest revision.
>
> > For all tested synthetic cases, can we compare the found affine mapping with the true one as requested by the editor in the previous round of reviews?
>
> Thank you for this suggestion; we have added this analysis. On the pure synthetic data settings (Figs S1, S6, and S8), ConDo outperforms the baselines in estimating the true parameters; and when sample size increases (S8), ConDo estimation improves. But on ANSUR (S10), we do not observe systematic benefits from ConDo in feature-space mapping parameter estimation.
>
> > Give an example of a commonly used domain adaptation benchmark where the confounding shift doesn't need to be defined artificially.
>
> The confounded shift in the SNAREseq single-cell multi-omics data (Section 4.7) is not artificial; this dataset has been frequently used to evaluate single-cell data alignment methods.

---

### Author Response · Authors · 2024-09-16
**Shared response to reviews**

We thank the reviewers for their careful attention. We appreciate their recognition of the importance of the problem and of the technical quality and elegance of our framework. We have individually responded to reviewer-specific comments. Below, we address reviewers' shared concerns around the current limitations of the ConDo framework.

We agree with reviewers' comments that extensions to nonlinear and/or nondeterministic mappings, extensions to handle hidden and/or partially-observed confounders, and theoretical analysis would all be valuable contributions. However, the fact that this work suggests multiple promising future research directions is a strength rather than a weakness of the paper, and is not a reason to reject from TMLR.

First, we note the two evaluation criteria of TMLR. "The acceptance decision for a submission is based on the answers to the following questions: Are the claims made in the submission supported by accurate, convincing and clear evidence? Would some individuals in TMLR's audience be interested in the findings of this paper? Papers should be accepted if they meet the criteria, even if the contribution or significance of the work is modest." The paper, despite its limitations, passes both criteria; the potential for additional contributions is not a reason for rejection. The paper only aims to "to begin to address this challenge" and describes the current limitations.

Second, as shown in experiments the current method is already useful (and therefore interesting) to many ML practitioners; we also note that ComBat's (Johnson et al., 2007) linear adaptation method has >7500 citations total and >900 in 2023. And we expect this to be interesting to other ML researchers who might contribute to overcoming the current limitations.

Third, we note that this paper already motivates and introduces a new problem setting, and proposes a new framework for it. This paper already contains much conceptual material, presenting a number of ideas that are already difficult to communicate in one place. Modifications to our setting and/or method may distract from the central idea of the paper, and are worthy of study in their own right.

---

### Decision · Action_Editor_GFLG · 2024-11-05

**Recommendation:** Accept with minor revision

**Comment:**

This paper has seen a major update since its previous submission and is in much better shape . All reviewers appreciated the proposed method and the response done by the authors. For all those reasons and the fact that the paper is solid and well justified, I recommend to accept the paper. But the since it remains sometimes hard to follow, some minor revisions are still needed before final acceptance.

More details below.

- Figure 1 is much better and the caption has been improved. But it should also reflect the notations of the paper in addition to concrete example of variables. The author should add to the figure what is X, Z and Y (that is not used in the paper but should be introduced) in the Figure so that the confounder (Z) and what is the final prediction task (prediction of Y from X) are clearly stated. As a matter of fact the author use Z for Y in Table 1 which is admittedly what traditional  DA method do but it makes the whole paper harder to follow because the proposed method use a confounder different from the label (since label is not available on target). This must be clarified in the final version because there is confusion for the reader between confounder and the actual label that is predicted on target using the estimated mapping. For the same reason the relation and assumption between the confounder an Y to have good performance also need to be discussed.

- For all numerical experiments with real data and related to the above point, the way the dataset is created (and split between source and target) is described in detail but not which feature is actually used as confounder. This is a problem for reproducibility and should be stated more clearly in each of the application subsections.

**Audience:**

This method will be of interest to some individuals in TMLR audience. Domain adaptation is a major challenge in ML and the proposed approach bring a novel way to address it using confounders.

**Claims And Evidence:**

The claims in the paper are supported by convincing evidence. Many numerical experiment show the interest of the method.

---

> ### Author Response · Authors · 2024-11-11
> **Revision notes**
>
> Thank you for attention and careful feedback. Our 2024-11-10 version of the manuscript contains the following revisions:
>
> Section 2: Updated Figure 1 and caption as requested.
>
> Section 3: Link to our de-anonymized Github repository.
>
> Section 3.1 (Our Assumption: Confounded Shift): We modify Table 1 as requested, and discuss "the relation and assumption between the confounder and Y" in a new paragraph, "Application to downstream prediction tasks".
>
> Section 4: Re "but not which feature is actually used as confounder", we modify as needed to contain these details, shown below. Also, our Github repository contains all our experimental scripts for ease of reproducibility.
>
> - Section 4.4 (ANSUR II): "We generated the source (and the target) dataset as a random subsample of 500 individuals with a 75%-25% (and a 25%-75%) male-female split, with gender as the confounding variable."
>
> - Section 4.5 (Image color): "...conditioning on each pixel label (a categorical confounder, taking on a value of either "water" or "sky")..."
>
> - Section 4.6 (California housing): "thus use LatLon coordinates as the two confounding variables"
>
> - Section 4.7 (SNARE-seq): "we perform adaptation controlling for cell type, the confounding variable"
>
> - Section 4.8 (Bladderbatch gene expression): "The confounder is 1d categorical (cancer or non-cancer)."